# Learning to Search Feasible and Infeasible Regions of Routing Problems with Flexible Neural k-Opt

**Yining Ma**
National University of Singapore
yiningma@u.nus.edu

**Zhiguang Cao**[*]
Singapore Management University
zgcao@smu.edu.sg

**Yeow Meng Chee**
National University of Singapore
ymchee@nus.edu.sg

## Abstract

In this paper, we present Neural k-Opt (NeuOpt), a novel learning-to-search (L2S) solver for routing problems. It learns to perform flexible k-opt exchanges based on a tailored action factorization method and a customized recurrent dual-stream decoder. As a pioneering work to circumvent the pure feasibility masking scheme and enable the autonomous exploration of both feasible and infeasible regions, we then propose the Guided Infeasible Region Exploration (GIRE) scheme, which supplements the NeuOpt policy network with feasibility-related features and leverages reward shaping to steer reinforcement learning more effectively. Additionally, we equip NeuOpt with Dynamic Data Augmentation (D2A) for more diverse searches during inference. Extensive experiments on the Traveling Salesman Problem (TSP) and Capacitated Vehicle Routing Problem (CVRP) demonstrate that our NeuOpt not only significantly outstrips existing (masking-based) L2S solvers, but also showcases superiority over the learning-to-construct (L2C) and learning-to-predict (L2P) solvers. Notably, we offer fresh perspectives on how neural solvers can handle VRP constraints. Our code is available: https://github.com/yining043/NeuOpt.

## 1 Introduction

Vehicle Routing Problems (VRPs), prevalent in various real-world applications, present NP-hard combinatorial challenges that necessitate efficient search algorithms [1, 2]. In recent years, significant progress has been made in developing deep (reinforcement) learning-based solvers (e.g., [3–11]), which automates the tedious algorithm design with minimal human intervention in a data-driven fashion. Impressively, these neural solvers developed over the past five years have closed the gap to or even surpassed some traditional hand-crafted solvers that have evolved for several decades [5, 12].

In general, neural methods for VRPs can be categorized into learning-to-construct (L2C), learning-to-search (L2S), and learning-to-predict (L2P) solvers, each offering unique advantages while suffering respective drawbacks. L2C solvers (e.g., [4, 13]) are recognised for their fast solution construction but may struggle to escape local optima. L2P solvers excel at predicting crucial information (e.g., edge heatmaps [14, 15]), thereby simplifying the search especially for large-scale instances, but may lack the generality to efficiently handle VRP constraints beyond the Traveling Salesman Problem (TSP). L2S solvers (e.g., [8, 9]) are designed to learn exploration in the search space directly; however, their search efficiency is still limited and lags behind the state-of-the-art L2C and L2P solvers. In this paper, we delve into the limitations of existing L2S solvers, and aim to unleash their full potential.

---

[*] Zhiguang Cao is the corresponding author.

37th Conference on Neural Information Processing Systems (NeurIPS 2023).

One potential issue of L2S solvers lies in their simplistic action space designs. Current L2S solvers that learn to control k-opt exchanges mostly leverage smaller $k$ values (2-opt [9, 16] or 3-opt [17]), partly because their models struggle to efficiently deal with larger $k$. To address this issue, we introduce Neural k-Opt (NeuOpt), a flexible L2S solver capable of handling k-opt for any $k \geq 2$. Specifically, it employs a tailored action factorization method that simplifies and decomposes a complex k-opt exchange into a sequence of basis moves (S-move, I-move, and E-move) with the number of I-moves determining the $k$ of a specifically executed k-opt action[1]. Such design allows k-opt exchanges to be easily constructed step-by-step, which more importantly, provides the deep model with the flexibility to explicitly and automatically determine an appropriate $k$. This further enables varying $k$ values to be combined across different search steps, striking a balance between coarse-grained (larger $k$) and fine-grained (smaller $k$) searches. Correspondingly, we design a Recurrent Dual-Stream (RDS) decoder to decode such action factorization, which consists of recurrent networks and two complementary decoding streams for contextual modeling and attention computation, thereby capturing the strong correlations and dependencies between removed and added edges.

Besides, existing L2S solvers confine the search space to feasible regions based on feasibility masking. By contrast, we introduce a novel Guided Infeasible Region Exploration (GIRE) scheme that promotes the exploration of both feasible and infeasible regions. GIRE enriches the policy network with additional features that indicate constraint violations in the current solution and the exploration behaviour statistics in the search space. It also includes reward shaping to regulate extreme exploration behaviours and incentivize exploration at the boundaries of feasible and infeasible regions. Our GIRE offers four advantages: 1) it circumvents the non-trivial calculation of ground-truth action masks, particularly beneficial for constrained VRPs or broader action space as in NeuOpt, 2) it fosters searches at the more promising feasibility boundaries, similar to traditional solvers [18–21], 3) it bridges (possibly isolated) feasible regions, helping escape local optima and discover shortcuts to better solutions (See Figure 2), and 4) it forces explicit awareness of the VRP constraints, facilitating the deep model to understand the problem landscapes. In this paper, we apply GIRE to the Capacitated Vehicle Routing Problem (CVRP), though we note that it is generic to most VRP constraints.

Moreover, our NeuOpt leverages a Dynamic Data Augmentation (D2A) method during inference to enhance the search diversity and escape local optima. Our NeuOpt is trained via the reinforcement learning (RL) algorithm tailored in our previous work [12]. Extensive experiments on classic VRP variants (TSP and CVRP) validate our designs and demonstrate the superiority of NeuOpt and GIRE over existing approaches. Our contributions are four-fold: 1) we present NeuOpt, the first L2S solver that is flexible to handle k-opt with any $k \geq 2$ based on a tailored formulation and a customized RDS decoder, 2) we introduce GIRE, the first scheme that extends beyond feasibility masking, enabling exploration of both feasible and infeasible regions in the search space, thereby bringing multiple benefits and offering fresh perspectives on handling VRP constraints, 3) we propose a simple yet effective D2A inference method for L2S solvers, and 4) we unleash the potential of L2S solvers and allow it to surpass L2C, L2P solvers, as well as the strong LKH-3 solver [20] on CVRP.

## 2    Literature review

We categorize recently developed neural methods for solving vehicle routing problems (VRPs) into *learning-to-construct* (L2C), *learning-to-search* (L2S), and *learning-to-predict* (L2P) solvers.

**L2C solvers.** They learn to construct solutions by iteratively adding nodes to the partial solution. The first modern L2C solver is Ptr-Net [22] based on a Recurrent Neural Network (RNN) and supervised learning (extended to RL in [23] and CVRP in [24]). The Graph Neural Networks (GNN) were then leveraged for graph embedding [25] and faster encoding [26]. Later, the Attention Model (AM) was proposed by Kool et al. [3], inspiring many subsequent works (e.g., [4, 27–31]), where we highlight Policy Optimization with Multiple Optima (POMO) [4] which significantly improved AM with diverse rollouts and data augmentations. The L2C solvers can produce high-quality solutions within seconds using greedy rollouts; however, they are prone to get trapped in local optima, even when equipped with post-hoc methods (e.g., sampling [3], beam search [28], etc), or, advanced strategies (e.g., invariant representation [13, 32], learning collaborative policies [33], etc). Recently, the Efficient Active Search (EAS) [5] addressed such issues by updating a small subset of pre-trained

---

[1]Broadly, in k-opt, added edges may coincide with removed ones. Thus, 2-opt may be viewed as degenerated 8-opt. To avoid ambiguity, unless specified, we refer to k-opt as an exchange introducing $k$ entirely new edges.

model parameters on each test instance, which could be further boosted if coupled with Simulation Guided Beam Search (SGBS) [34], achieving the current state-of-the-art performance for L2C solvers.

**L2S solvers.** They learn to iteratively refine a solution to a new one, featuring a search process. Early attempts, e.g., NeuRewriter [35] and L2I [36], relied heavily on traditional local search algorithms and long run time. The NLNS solver [8] improved upon them by controlling a ruin-and-repair process that destroys parts of the solution using handcrafted operators and then fixes them using a learned deep model. Besides, the crossover exchanges between solutions were also learned in [37]. Recently, several L2S solvers focused on controlling the k-opt heuristic that is more suitable for VRPs [20, 38]. Wu et al. [39] made an early attempt to guide 2-opt, showing superior performance than the L2C solver AM [3]. Ma et al. [9] improved Wu et al. [39] by replacing vanilla attention with Dual-Aspect Collaborative Attention (DAC-Att) and a cyclic positional encoding method. The DAC-Att was then upgraded to Synthesis Attention (Synth-Att) [12] to reduce computational costs. Besides, Costa et al. [16] proposed an RNN-based policy to govern 2-opt, which was extended to 3-opt in [17] with higher efficiency. However, these neural k-opt solvers are limited by a small and fixed $k$. Besides, although L2S solvers strive to surpass L2C solvers by directly learning to search, they are still inferior to those state-of-the-art L2C solvers (e.g., POMO [4] and EAS [5]) even when given prolonged run time.

**L2P solvers.** They learn to guide the search by predicting critical information. Joshi et al. [14] proposed using GNN models to predict heatmaps that indicate probabilities of the presence of an edge in the optimal solution, which then uses beam search to solve TSP. It was leveraged for larger-scale TSP instances in [6] based on divide-and-conquer, heatmap merging, and Monte Carlo Tree Search. In the GLS solver [40], a similar GNN was used to guide the traditional local search heuristics. More recently, the DIFUSCO solver [15] proposed to replace those GNN models with diffusion models [41]. Compared to L2C or L2S solvers, L2P solvers exhibit better scalability for large instances; however, they are mostly limited to supervised learning and TSP only, due to challenges in preparing training data and the ineffectiveness of heatmaps in handling VRP constraints. Though L2P solver DPDP [42] managed to solve CVRP with dynamic programming, it was outstripped by L2C solver EAS[5]. Recently, L2P solvers also explored predicting a latent continuous space for the underlying discrete solution space, where the latent space is then searched using differential evolution in [43] or gradient optimizer in [7]. Still, they can be either time-consuming or ineffective in tackling VRP constraints.

**Feasibility satisfaction.** Most neural solvers handle VRP constraints using the masking scheme that filters out invalid actions (e.g., [12, 44]). However, few works considered better ways of constraint handling. Although the works [45, 46] attempted to use mask prediction loss to enhance constraint awareness, they overlooked the benefits of the temporary constraint violation applied in many traditional solvers [18–21]. Lastly, we note that constraint handling in VRPs largely differs from safe RL tasks that focus on fully avoiding risky actions in uncertain environments [47, 48].

## 3 Preliminaries and notations

**VRP notations.** VRP aims to minimize the total travel cost (tour length) while serving a group of customers subject to certain constraints. It is defined on a complete directed graph $\mathcal{G} = \{\mathcal{V}, \mathcal{E}\}$ where $x_i \in \mathcal{V}$ are nodes (customers) and $e(x_i \to x_j) \in \mathcal{E}$ are possible edges (route) weighted by the Euclidean distance between $x_i$ and $x_j$. In the Traveling Salesman Problem (TSP), the solution is a Hamiltonian cycle that visits each node exactly once and returns to the starting one. In the Capacitated Vehicle Routing Problem (CVRP), a depot $x_0$ is added to $\mathcal{V}$, and each customer node $x_i (i \geq 1)$ is assigned a demand $\delta_i$. A CVRP solution consists of multiple sub-tours, each representing a vehicle departing from the depot, serving a subset of customers, and returning to the depot, where each $x_i (i \geq 1)$ is visited exactly once and the total demand of a sub-tour must not exceed the vehicle capacity $\Delta$. For instance, $\tau = \{x_0 \to x_2 \to x_1 \to x_0 \to x_3 \to x_0\}$ is a CVRP-3 solution with $\delta = [5, 5, 9]$ and $\Delta = 10$.

**Traditional k-opt heuristic.** The k-opt heuristic iteratively refines a given solution by exchanging $k$ existing edges with $k$ (entirely) new ones. The Lin-Kernighan (LK) algorithm [49] narrowed the search with several criteria, where we underscore the *sequential exchange criterion*. It requires: for each $i = 1, \ldots, k$, the removed edge $e_i^{\text{out}}$ and added edge $e_i^{\text{in}}$ must share an endpoint, and so must $e_i^{\text{in}}$ and $e_{i+1}^{\text{out}}$. This allows a simplified sequential search process, alternating between removing and adding edges. Moreover, the LK algorithm considers scheduling varying $k$ values in a repeated ascending order, so as to escape local optima by varying search neighbourhoods. Inspired by them, our paper proposes a powerful L2S solver that performs flexible k-opt exchanges with automatically chosen $k$

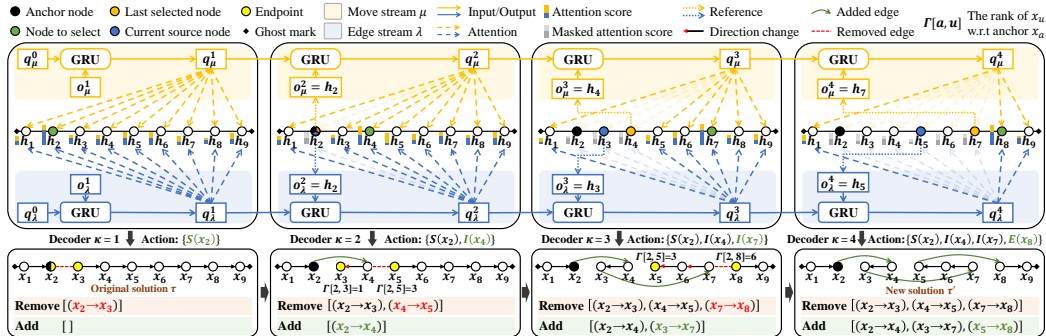

Figure 1: Illustration of using our RDS decoder to determine a 3-opt exchange on TSP-9 given $K = 4$ steps (with the E-move chosen at the final decoding step). The upper portion depicts a visual representation of how the dual-stream attentions (move stream $\mu$ and edge stream $\lambda$) are computed. The lower portion demonstrates how the inferred basis moves lead to the modification of the current solution. At step $\kappa$, RDS computes dual-stream attention from representations of historical decisions $q_\mu^\kappa$, $q_\lambda^\kappa$ to node embeddings $h_i$, thereby deciding a basis move $\Phi_\kappa(x_\kappa)$ by selecting $x_\kappa$. Ghost marks indicate the same location of a cyclic solution when viewed in a flat perspective as in this figure.

based on our action factorization method and the customized decoder. Recently, the LK algorithm has been implemented by Helsgaun [50] in the open-source LKH solver, with additional non-sequential exchanges and an edge candidate set. It was then upgraded to LKH-2 [51], leveraging more general k-opt exchanges, divide-and-conquer strategies, etc. The latest release, LKH-3 [20], further tackled constrained VRPs by penalizing constraint violations, making it a generic and powerful solver that serves as a benchmark for neural methods. Finally, we note that k-opt has been a foundation for various solvers, including the state-of-the-art Hybrid Genetic Search (HGS) solver [21] for CVRP.

## 4   Neural k-opt (NeuOpt)

Designing an effective neural k-opt solver necessitates addressing challenges potentially overlooked in prior works. Firstly, the solver should be generic for any given $k \geq 2$, using a unified formulation and architecture. Secondly, it should coherently parameterize the complex action space while accounting for the strong correlations and dependencies between removed and added edges. Finally, it should dynamically adjust $k$ to balance coarse-grained (larger $k$) and fine-grained (smaller $k$) search steps.

In light of them, we introduce Neural k-Opt (NeuOpt) in this section. We first present our action factorization method for flexible k-opt exchanges, and then demonstrate our decoder to parameterize such actions, followed by the dynamic data augmentation method for inference. Note that this section focuses on TSP only, and we extend our NeuOpt to handle other VRP constraints in Section 5.

### 4.1   Formulations

We introduce a new factorization method that constructs a k-opt exchange using a combination of three *basis moves*, namely the *starting move*, the *intermediate move*, and the *ending move*. Concretely, the sequential k-opt can be simplified as performing one S-move, several (possibly none) I-moves, and finally one E-move, where the choice of the $k$ corresponds to determining the number of I-moves.

**S-move.** The starting move removes an edge $e^{\text{out}}(x_a \rightarrow x_b)$, converting a TSP solution $\tau$ (a Hamiltonian cycle) into an open Hamiltonian path with two endpoints $x_a$ and $x_b$. It is executed only at the beginning of action construction. We denote S-move as $S(x_a)$ since $x_b$ can be uniquely determined if $x_a$ is specified. We term the source node $x_a$ as *anchor node* to compute the *node rank*.

**Definition 1 (Node rank w.r.t. anchor node)** *Given an instance $\mathcal{G} = (\mathcal{V}, \mathcal{E})$ and a solution $\tau$, let $x_a$ be the anchor node, as specified in an S-move. The node rank of $x_u$ ($x_u \in \mathcal{V}$) w.r.t. $x_a$, denoted by $\Gamma[x_a, x_u]$ or $\Gamma[a, u]$, is defined as the minimal number of edges in $\tau$ needed to reach $x_u$ from $x_a$.*

**I-move.** The intermediate move adds a new edge, removes an existing edge, and fixes edge directions, transforming the open Hamiltonian path into a new one. Let $x_i, x_j$ be the endpoints of the Hamiltonian

path before the I-move (with $\Gamma[a,i] < \Gamma[a,j]$), and let $e^{\text{in}}(x_u \to x_v)$ be the introduced new edge. To avoid conflicts between consecutive I-moves, we impose sequential conditions on $e^{\text{in}}$: (1) its source node $x_u$ must be the endpoint of the current Hamiltonian path with a lower node rank, i.e., $x_u = x_i$, and (2) the node rank of its target node $x_v$, i.e., $\Gamma[a,v]$, must be higher than the node ranks of the two current endpoints $x_i, x_j$, i.e., $\Gamma[a,i] < \Gamma[a,j] < \Gamma[a,v]$. The removal of edge $e^{out}(x_v \to x_w)$ followed when $x_v$ is chosen, and directions of the edges between $x_j$ and $x_v$ are reversed to yield a valid Hamiltonian path. Since an I-move can be uniquely determined by $x_v$, we denote it as $I(x_v)$.

**E-move.** The ending move adds a new edge connecting the two endpoints of the current Hamiltonian path, converting it into a Hamiltonian cycle, i.e., a new TSP solution $\tau'$. It is executed only at the end of action construction. Since it is uniquely determined without specifying any node, we denote it as $E(x_{\text{null}})$. Note that if we relax the condition (2) of I-move to $\Gamma[a,j] \le \Gamma[a,v]$, an E-move can be treated as a *general I-move* that selects $x_v = x_j$, denoted as $I'(x_j)$ or $E(x_j)$.

**MDP formulations.** The examples of using the above *basis moves* to factorize 1-opt (void action), 2-opt, 3-opt, and 4-opt are depicted in Appendix A. Next, we present the Markov Decision Process (MDP) formulation $\mathcal{M} = (\mathcal{S}, \mathcal{A}, \mathcal{T}, \mathcal{R}, \gamma < 1)$ for our NeuOpt. At step $t$, the **state** $s_t = \{\mathcal{G}, \tau_t, \tau_t^{\text{bsf}}\}$ describes the current instance $\mathcal{G}$, the current solution $\tau_t$, and the best-so-far solution $\tau_t^{\text{bsf}}$ found before step $t$. Given a maximum number of allowed *basis moves* $K(K \ge 2)$, an **action** consists of $K$ *basis moves* $\Phi_\kappa(x_\kappa)$, i.e., $a_t = \{\Phi_\kappa(x_\kappa), \kappa = 1, \ldots, K\}$, where the first move $\Phi_1(x_1) = S(x_1)$ is an S-move, and the rest is either an I-move or an E-move (in the form of general I-move). Note that 1) we permit the I-move as the last move, adding an E-move during state transition if so, and 2) to ensure the action length is always $K$, we include null actions if E-move early terminates the action. Our **state transition** rule $\mathcal{T}$ is deterministic and it updates $s_t$ to $s_{t+1}$ based on the above action factorization method. We use **reward** $r_t = f(\tau_t^{\text{bsf}}) - min\left[f(\tau_{t+1}), f(\tau_t^{\text{bsf}})\right]$ following [9, 12].

### 4.2 Recurrent Dual-Stream (RDS) decoder

Our NeuOpt adopts an encoder-decoder-styled policy network. We use the encoder from our previous work [12] but upgrade the linear projection to an MLP for embedding generation (details are presented in Appendix B). Here, we introduce the proposed recurrent dual-stream (RDS) decoder to effectively parameterize the aforementioned k-opt action factorization (see Figure 1 for a concrete example).

**GRUs for action factorization.** An action $a = \{\Phi_\kappa(x_\kappa), \kappa = 1, \ldots, K\}$ is constructed sequentially by $K$ steps, where each decoding step $\kappa$ specifies a basis move type $\Phi_\kappa$ and a node $x_\kappa$ to instantiate the move. Such decoding can be further simplified to a *node selection process*, with move types inferred based on: (1) for $\kappa = 1$, it is an S-move; (2) for $\kappa > 1$, it is an I-move if $\Gamma[a,j_\kappa] < \Gamma[a,\kappa]$ (assuming $x_{i_\kappa}, x_{j_\kappa}$ are the Hamiltonian path endpoints before the $\kappa$-th move with $\Gamma[a,i_\kappa] < \Gamma[a,j_\kappa]$), otherwise, if $x_\kappa = x_{j_\kappa}$, it is an E-move that early stops the decoding. Formally, we use factorization:

$$\pi_\theta(a|s) = P_\theta(\Phi_1(x_1), \Phi_2(x_2), \ldots, \Phi_K(x_K)|s) = \prod_{\kappa=1}^{K} P_\theta^\kappa(\Phi_\kappa|\Phi_1, \ldots, \Phi_{\kappa-1}, s) \qquad (1)$$

where $P_\theta^\kappa$ is a categorical distribution over $N$ nodes for node selection. Our decoder leverages the Gated Recurrent Units (GRUs) [52] to help parameterize the conditional probabilities $P_\theta^\kappa$. Given node embeddings $h \in \mathbb{R}^{N \times d}$ from the encoders ($h_i$ is a $d$-dimensional vector), the decoder first computes hidden representations $q^\kappa$ to model the historical move decisions $\{\Phi_1, \ldots, \Phi_{k-1}\}$ (the conditions of $P_\theta^\kappa$) using Eq. (2), where $q^\kappa$ (hidden state of GRU) is derived from $q^{\kappa-1}$, based on an input $o^\kappa$. For better contextual modeling, we consider two streams, $\mu$ and $\lambda$, which differ from each other by learning independent parameters and taking different inputs $o^\kappa$ at each $\kappa$. The $q_\mu^0 = q_\lambda^0 = \frac{1}{N}\sum_{i=1}^{N} h_i$ are initialized by the mean pooling of node embeddings (i.e., a graph embedding).

$$q_\mu^\kappa = \text{GRU}\left(o_\mu^\kappa, q_\mu^{\kappa-1}\right), \; q_\lambda^\kappa = \text{GRU}\left(o_\lambda^\kappa, q_\lambda^{\kappa-1}\right) \qquad (2)$$

**Dual-stream contextual modeling.** We employ a *move stream* $\mu$ and an *edge stream* $\lambda$ during contextual modeling and subsequent attention computation. On the one hand, to provide a relatively overall view, the *move stream* $\mu$ considers modeling historical move decisions by taking the node embedding of the last selected node $x_{\kappa-1}$, which is a representation of the past selected move $\Phi_{\kappa-1}(x_{\kappa-1})$, as its GRU input, i.e., $o_\mu^\kappa = h_{\kappa-1}$. On the other hand, to offer a relatively detailed view, the *edge stream* $\lambda$ focuses on edge proposals in each step $\kappa$ by taking the node embedding of $x_{i_\kappa}$, which is stipulated to be the source node of the edge to be introduced in step $\kappa$, as its GRU input,

i.e., $o_\lambda^\kappa = h_{i_\kappa}$. We intend the two streams to complement each other, coherently capturing complex correlations and dependencies among historical decisions. For the initialization, we use learnable parameters $o_\mu^1 = \hat{o}$, $o_\lambda^1 = \tilde{o}$. Based on the attention between $q_\mu, q_\lambda$ and node embeddings $h$, the node selection scores suggested by both streams can then be calculated:

$$
\begin{aligned}
\mu_\kappa &= \text{Tanh}\left( (q_\mu^\kappa W_\mu^{\text{Query}} + h W_\mu^{\text{Key}}) + (q_\mu^\kappa W_\mu^{\text{Query}'}) \odot (h W_\mu^{\text{Key}'}) \right) W_\mu^O, \\
\lambda_\kappa &= \text{Tanh}\left( (q_\lambda^\kappa W_\lambda^{\text{Query}} + h W_\lambda^{\text{Key}}) + (q_\lambda^\kappa W_\lambda^{\text{Query}'}) \odot (h W_\lambda^{\text{Key}'}) \right) W_\lambda^O,
\end{aligned}
\tag{3}
$$

where all matrices $W \in \mathbb{R}^{d \times d}$ are trainable parameters and we consider both summation and Hadamard products. Afterwards, the final categorical distribution $P_\theta^\kappa$ is induced by: $\text{Softmax}(C \cdot \tanh(\mu_\kappa + \lambda_\kappa))$ where $C = 6$ and invalid choices that do not satisfy $\Gamma[a, j_\kappa] \leq \Gamma[a, \kappa]$ are masked as $-\infty$ (an E-move is enforced if all nodes are masked, which happens when the highest-ranked node was chosen in the previous I-move). A node $x_\kappa$ is then sampled from $P_\theta^\kappa$, after which its basis move type $\Phi_\kappa$ can be inferred. The decoding process continues until it early terminates or reaches the maximum limit $K$.

### 4.3 Inference with the dynamic data augmentation (D2A)

Our NeuOpt is trained using the RL algorithm tailored in our previous work [12]. During inference, our previous work [12] employed a data augmentation method that applies optimal-solution-invariant transformations (e.g., flipping coordinates) on an instance, $\mathcal{G}$, so as to generate a set of instances $\{\mathcal{G}_1, \mathcal{G}_2, ..., \}$ for the trained model to solve. However, it was performed only once at the beginning of the inference. In this paper, we propose Dynamic Data Augmentation (D2A), an enhancement that generates new augmented instances each time the solver fails to find a better solution (i.e., getting trapped in local optima) within a consecutive maximum of $T_{\text{D2A}}$ steps. This allows for more diverse searches. Further details regarding training and inference algorithms are provided in Appendix C.

## 5 Guided infeasible region exploration (GIRE)

As demonstrated in Figure 2, the feasible region is usually fragmented with isolated islands. Masking in neural solvers typically restricts the search to feasible regions, which may lead to inefficient search trajectories or failure to find global optima. Our GIRE explores both feasible and infeasible regions, fostering shortcut discovery, more promising boundary searches, and the identification of possibly isolated regions. We now illustrate our GIRE by handling the capacity constraint in CVRP.

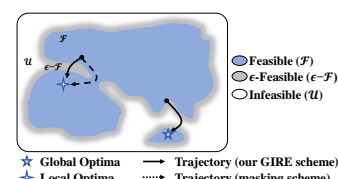

Figure 2: A search space example.

**Definition 2 (Search space of CVRP)**: *We define the feasible search space $\mathcal{F}$ as the set of solutions that satisfy both TSP and the (CVRP-specific) capacity constraint; the infeasible search space $\mathcal{U}$ as the set of solutions that satisfy the TSP constraint but not the capacity constraint; and the $\epsilon$-feasible search space $\epsilon\text{-}\mathcal{F} \subseteq \mathcal{U}$ as the set of solutions with a capacity violation percentage not exceeding $\epsilon$.*

**Feature supplement.** GIRE suggests supplementing *Violation Indicator* (VI) features and *Exploration Statistics* (ES) features into the policy network to identify specific constraint violations and understand its ongoing exploration behaviour. The **VI features** flag the specific infeasible portions of the current solution. For CVRP, we use two binary variables indicating if the cumulative demands exceed the capacity before or after visiting a particular node, respectively, which are treated as node features during the node embedding generation. Such an idea could work with most VRP constraints, e.g., indicating the nodes that violate their time windows (TW) for CVRP-TW. The **ES features** provide the network with ongoing exploration behaviour statistics to guide future exploration behaviour. We define $\mathcal{H}_t := \{(\tau_{t'} \to \tau_{t'+1})\}|_{t'=t-T_{\text{his}}}^{t-1}$ as the collection of the most recent $T_{\text{his}}$ steps of solution transition record (if they exist). The ES features $\mathcal{J}_t$ consist of estimated feasibility transition probabilities derived from $\mathcal{H}_t$, including $P(\tau \in \mathcal{F}, \tau' \in \mathcal{U})$, $P(\tau \in \mathcal{U}, \tau' \in \mathcal{F})$, $P(\tau \in \mathcal{F}, \tau' \in \mathcal{F})$, $P(\tau \in \mathcal{U}, \tau' \in \mathcal{U})$, $P(\tau' \in \mathcal{F}|\tau \in \mathcal{U})$, $P(\tau' \in \mathcal{U}|\tau \in \mathcal{F})$, $P(\tau' \in \mathcal{F}|\tau \in \mathcal{F})$, and $P(\tau' \in \mathcal{U}|\tau \in \mathcal{U})$, along with a binary indicator w.r.t the feasibility of the current solution $\tau_t$. In order to make the policy network decisions dependent on these ES features, we introduce two hypernetworks [53], namely, $\text{MLP}_\mu$ and $\text{MLP}_\lambda$, which take ES features $\mathcal{J}_t$ as inputs and generate the parameters of the last decoder layer, i.e., $W_\mu^O$, $W_\lambda^O$ of Eq. (3), respectively. We adopt the structure $(9 \times 8 \times d)$, and share the first hidden layer between $W_\mu^O$ and $W_\lambda^O$ to reduce computational costs.

**Reward shaping.** Moreover, GIRE employs reward shaping:

$$r_t^{\text{GIRE}} = r_t + \alpha \cdot r_t^{\text{reg}} + \beta \cdot r_t^{\text{bonus}} \qquad (4)$$

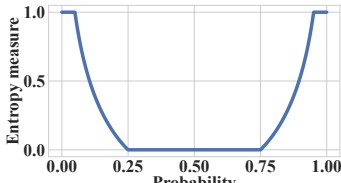

Figure 3: Plot of $\mathbb{H}[P]$.

to guide the reinforcement learning, where $r_t$ is the original reward, $r_t^{\text{reg}}$ regulates extreme exploration behaviours; $r_t^{\text{bonus}}$ encourages exploration in the $\epsilon$-feasible regions; and $\alpha$ and $\beta$ are reward shaping weights (we use 0.05 for both). Here, the regulation $r_t^{\text{reg}}$ is determined by an entropy measure $\mathbb{H}[P]$ of the estimated conditional transforming probabilities $P_t(\mathcal{U}|\mathcal{U}) = P(\tau' \in \mathcal{U} | \tau \in \mathcal{U})$ and $P_t(\mathcal{F}|\mathcal{F}) = P(\tau' \in \mathcal{F} | \tau \in \mathcal{F})$:

$$r_t^{\text{reg}} = -\mathbb{E}[r_t] \times [\mathbb{H}[P_t(\mathcal{U}|\mathcal{U})] + \mathbb{H}[P_t(\mathcal{F}|\mathcal{F})]],$$
$$\mathbb{H}[P] = \text{Clip}\{1 - c_1 \log_2[c_2 \pi e P(1-P)], 0, 1\}, c_1 = 0.5, c_2 = 2.5, \qquad (5)$$

where expectation $\mathbb{E}[r_t]$, that suggests the magnitude of $r_t^{\text{reg}}$, is estimated during training; the entropy measure $\mathbb{H}[P]$, as shown in Figure 3, imposes larger penalties when $P$ is either too high or too low (indicating extreme exploration behaviour). The bonus $r_t^{\text{bonus}}$ utilizes a similar reward function as the regular reward $r_t$; however, it only considers an infeasible but $\epsilon$-feasible solution as a potential new best-so-far solution. More illustrations and discussions on GIRE designs are detailed in Appendix D.

## 6 Experiments

We conduct experiments on TSP and CVRP, with sizes $N = 20, 50, 100$ following the conventions [9, 13, 37]. Training and test instances are uniformly generated following [3]. For NeuOpt, we use $K = 4$; the initial solutions are sequentially constructed in a *random* fashion for both training and inference. Results were collected using a machine equipped with NVIDIA 2080TI GPU cards and an Intel E5-2680 CPU at 2.40GHz. More hyper-parameter details, discussions, and additional results are available in Appendix E. Our PyTorch code and pre-trained models are publicly available[2].

### 6.1 Comparison studies

**Setup.** In Table 1, we benchmark our NeuOpt (TSP) and NeuOpt-GIRE (CVRP) against a variety of neural solvers, namely, **1) L2P solvers**: *GCN+BS* [14] (TSP only), *Att-GCN+MCTS* [6] (TSP only), *GNN+GLS* [40] (TSP only), *CVAE-Opt-DE* [43], *DPDP* [42] (state-of-the-art), *DIMES* [7] (TSP only), *DIFUSCO* [15] (TSP only, state-of-the-art); **2) L2C solvers**: *AM+LCP* [33], *POMO* [4], *Pointerformer* [32] (TSP only), *Sym-NCO* [13], *POMO+EAS* [5], *POMO+EAS+SGBS* [34] (state-of-the-art); and **3) L2S solvers**: *Costa et al.* [16] (TSP only), *Sui et al.* [17] (TSP only), *Wu et al.* [39], *NLNS* [8] (CVRP only), *NCE* [37] (CVRP only), *DACT* [9] (state-of-the-art). To ensure fairness, we test their publicly available pre-trained models on our hardware and test datasets. Those marked with ‡ are sourced from their original papers due to difficulties in reproducing, among which we find potential issues marked with #. More implementation details are listed in Appendix E. Following the conventions [5, 9, 34], we report the metrics of objective values and optimality gaps averaged on a test dataset with 10k instances, where the total run time is measured under the premise of using one GPU for neural methods and one CPU for traditional ones. The gaps are computed w.r.t. the exact solver *Concorde* [54] for TSP and the state-of-the-art traditional solver *HGS* [21] for CVRP. We also include the *LKH* [20, 51] as baselines. However, we note that it is hard to be absolutely fair when comparing the run time between those CPU-based traditional solvers and GPU-based neural solvers. The baselines are grouped, where the last group comprises variations of our NeuOpt, differentiated by the number of augments (marked as 'D2A=') and the number of inference steps (marked as 'T=').

**TSP results.** Compared to **L2P solvers**, NeuOpt (D2A=1, T=1k) surpasses GCN+BS, CVAE-Opt-DE, and GNN+GLS in all problem sizes with shorter run time. With increased T, NeuOpt continues reducing the gaps, and outshines the state-of-the-art DIFUSCO solver with less time at T=10k steps. The NeuOpt (D2A=5, T=1k), with more augmentations, shows lower gaps than NeuOpt (D2A=1, T=5k) in the same solution search count, where it achieves the lowest gap of 0.00% at (D2A=5, T=5k) on all sizes. Despite the longer run time compared to DPDP and Att-GCN+MCTS, their high efficiency is limited to TSP, and our NeuOpt could be potentially boosted by leveraging heatmaps

[2]https://github.com/yining043/NeuOpt

Table 1: Performance comparison of NeuOpt with various baselines on TSP and CVRP benchmarks.

| Problem | Method | Model Type | Post (Per-Ins.) Proc. | $N=20$ Obj.↓ | $N=20$ Gap↓ | $N=20$ Time↓ | $N=50$ Obj.↓ | $N=50$ Gap↓ | $N=50$ Time↓ | $N=100$ Obj.↓ | $N=100$ Gap↓ | $N=100$ Time↓ |
|---|---|---|---|---|---|---|---|---|---|---|---|---|
| TSP | Concorde [54] | Exact | - | 3.827 | - | 2m | 5.696 | - | 9m | 7.765 | - | 43m |
| | LKH-2 [51] | H | - | 3.827 | 0.00% | 6m | 5.696 | 0.00% | 1.3h | 7.765 | 0.00% | 5.7h |
| | GCN+BS [14][#] | L2P/SL | BS+H | 3.827 | 0.00% | 15m | 5.698 | 0.04% | 23m | 7.869 | 1.35% | 46m |
| | Att-GCN+MCTS [6][‡,#] | L2P/SL | MCTS | (≈3.830) | (≈0.00%) | ≈2m | (≈5.691) | (≈0.01%) | ≈8m | (≈7.764) | (≈0.04%) | ≈15m |
| | GNN+GLS [40] (relocate+2-opt)[‡] | L2P/SL | GLS | - | ≈0.00% | ≈2.8h | - | ≈0.00% | ≈2.8h | - | ≈0.58% | ≈2.8h |
| | CVAE-Opt-DE [43][‡] | L2P/UL | DE | - | ≈0.00% | ≈1.2d | - | ≈0.02% | ≈2.5d | - | ≈0.34% | ≈1.8d |
| | DPDP [42] (100k) | L2P/SL | DP | | - | | | - | | 7.765 | 0.00% | 1.9h |
| | DIMES [7] (T=10)[‡,#] | L2P/RL | AS+M+M | | - | | | - | | (≈7.762) | (≈0.01%) | - |
| | DIFUSCO [15] (T=50, S=16)[#] | L2P/SL | 2-opt | | - | | 5.696 | 0.01% | 5.8h | 7.766 | 0.02% | 21.7h |
| | AM+LCP* [33] ({1280, 45}) | L2C/RL | - | 3.828 | 0.01% | 2.1h | 5.699 | 0.05% | 4.9h | 7.811 | 0.60% | 10.9h |
| | Pointerformer [32] (A=8, T=200) | L2C/RL | - | 3.827 | 0.00% | 13m | 5.697 | 0.02% | 1.1h | 7.773 | 0.11% | 5.6h |
| | Sym-NCO [13] (A=8, T=200) | L2C/RL | - | | - | | | - | | 7.771 | 0.08% | 5.6h |
| | POMO [4] (A=8, T=200) | L2C/RL | - | 3.827 | 0.00% | 13m | 5.696 | 0.00% | 1.1h | 7.770 | 0.07% | 5.6h |
| | POMO+EAS [5] (A=8, T=200) | L2C/RL | AS | 3.827 | 0.00% | 24m | 5.696 | 0.00% | 2h | 7.769 | 0.05% | 10.9h |
| | POMO+EAS+SGBS [34] (short) | L2C/RL | AS+BS | | - | | | - | | 7.767 | 0.04% | 6.5h |
| | POMO+EAS+SGBS [34] (long) | L2C/RL | AS+BS | | - | | | - | | 7.767 | 0.03% | 1.1d |
| | Costa et al. [16] (2-opt, T=2k) | L2S/RL | - | 3.827 | 0.00% | 31m | 5.703 | 0.12% | 40m | 7.824 | 0.77% | 1.1h |
| | Sui et al. [17] (3-opt, T=2k)[‡] | L2S/RL | - | ≈3.84 | ≈0.00% | ≈32m | ≈5.70 | ≈0.08% | ≈48m | ≈7.82 | ≈0.74% | ≈1.3h |
| | Wu et al. [39] (2-opt, T=5k) | L2S/RL | - | | - | | 5.709 | 0.23% | 1.3h | 7.884 | 1.54% | 2h |
| | DACT [9] (2-opt, A=4, T=10k) | L2S/RL | - | 3.827 | 0.00% | 1.5h | 5.696 | 0.00% | 4.1h | 7.772 | 0.10% | 13.5h |
| | NeuOpt (D2A=1, T=1k) | L2S/RL | - | 3.827 | 0.00% | 2m | 5.697 | 0.02% | 6m | 7.790 | 0.33% | 17m |
| | NeuOpt (D2A=1, T=5k) | L2S/RL | - | 3.827 | 0.00% | 12m | 5.696 | 0.00% | 32m | 7.768 | 0.05% | 1.4h |
| | NeuOpt (D2A=1, T=10k) | L2S/RL | - | 3.827 | 0.00% | 23m | 5.696 | 0.00% | 1.1h | 7.766 | 0.02% | 2.8h |
| | NeuOpt (D2A=5, T=1k) | L2S/RL | - | 3.827 | 0.00% | 12m | 5.696 | 0.00% | 32m | 7.767 | 0.04% | 1.4h |
| | NeuOpt (D2A=5, T=3k) | L2S/RL | - | 3.827 | 0.00% | 35m | 5.696 | 0.00% | 1.6h | 7.765 | 0.01% | 4.2h |
| | NeuOpt (D2A=5, T=5k) | L2S/RL | - | 3.827 | 0.00% | 1h | 5.696 | 0.00% | 2.7h | 7.765 | 0.00% | 7h |
| CVRP | HGS [21] | H | - | 6.130 | - | 10.7h | 10.366 | - | 1.2d | 15.563 | - | 2.5d |
| | LKH-3 [20] | H | - | 6.135 | 0.08% | 17.9h | 10.375 | 0.09% | 2.8d | 15.647 | 0.54% | 5.7d |
| | CVAE-Opt-DE [43][‡] | L2P/UL | DE | ≈6.14 | - | ≈2.4d | ≈10.40 | - | ≈4.7d | ≈15.75 | - | ≈11d |
| | DPDP [42] (1000k) | L2P/SL | DP | | - | | | - | | 15.627 | 0.41% | 1.2d |
| | AM+LCP [33] ({2560, 1})[‡] | L2C/RL | - | ≈6.15 | ≈0.33% | ≈23m | ≈10.52 | ≈1.48% | ≈52m | ≈16.00 | ≈2.81% | ≈2.1h |
| | Sym-NCO [13] (A=8, T=200) | L2C/RL | - | | - | | | - | | 15.702 | 0.89% | 7.2h |
| | POMO [4] (A=8, T=200) | L2C/RL | - | 6.136 | 0.09% | 11m | 10.397 | 0.30% | 1.4h | 15.672 | 0.70% | 7.2h |
| | POMO+EAS [5] (A=8, T=200) | L2C/RL | AS | 6.132 | 0.04% | 38m | 10.379 | 0.13% | 3.1h | 15.610 | 0.30% | 16h |
| | POMO+EAS+SGBS [34] (short) | L2C/RL | AS+BS | | - | | | - | | 15.587 | 0.15% | 1d |
| | POMO+EAS+SGBS [34] (long) | L2C/RL | AS+BS | | - | | | - | | 15.579 | 0.10% | 4.1d |
| | NLNS [8] (Ruin-Repair, T=5k) | L2S/RL | - | 6.175 | 0.73% | 48m | 10.506 | 1.35% | 1.4h | 15.915 | 2.26% | 2.4h |
| | NCE [37] (CROSS exchange)[‡] | L2S/SL | - | ≈6.13 | ≈0.00% | ≈11h | ≈10.41 | ≈0.42% | ≈2.3d | ≈15.81 | ≈1.59% | ≈10.4d |
| | Wu et al. [39] (2-opt, T=5k) | L2S/RL | - | | - | | 10.544 | 1.72% | 4.2h | 16.165 | 3.87% | 5h |
| | DACT [9] (2-opt, A=6, T=10k) | L2S/RL | - | 6.130 | 0.01% | 4h | 10.383 | 0.16% | 16h | 15.736 | 1.11% | 1.7d |
| | NeuOpt-GIRE (D2A=1, T=1k) | L2S/RL | - | 6.132 | 0.03% | 4m | 10.430 | 0.61% | 12m | 15.865 | 1.94% | 28m |
| | NeuOpt-GIRE (D2A=1, T=5k) | L2S/RL | - | 6.130 | 0.00% | 20m | 10.382 | 0.16% | 59m | 15.698 | 0.87% | 2.3h |
| | NeuOpt-GIRE (D2A=1, T=10k) | L2S/RL | - | 6.130 | 0.00% | 41m | 10.375 | 0.08% | 2h | 15.656 | 0.60% | 4.6h |
| | NeuOpt-GIRE (D2A=5, T=6k) | L2S/RL | - | 6.130 | 0.00% | 2.1h | 10.369 | 0.03% | 5.9h | 15.610 | 0.30% | 13.8h |
| | NeuOpt-GIRE (D2A=5, T=20k) | L2S/RL | - | 6.130 | 0.00% | 6.8h | 10.367 | 0.01% | 19.7h | 15.586 | 0.15% | 1.9d |
| | NeuOpt-GIRE (D2A=5, T=40k) | L2S/RL | - | 6.130 | 0.00% | 13.7h | 10.367 | 0.01% | 1.6d | 15.579 | 0.10% | 3.8d |

[#] We found issues with their code, causing an underestimation of their reported gaps by roughly 0.02% (TSP-50) and 0.04% (TSP-100). Thus our reproduced gaps may vary from those in their papers. For not reproduced ones, we use (·) to indicate the underestimated values sourced directly from their papers.
[‡] The results with '≈' are based on their original papers since their code is either i) not publicly available (e.g., AM+CLP for CVRP), ii) raises (incompatible) errors that are difficult to fix on our hardware (e.g., Att-GCN+MCTS), or iii) requires long computation time to reproduce (e.g., CVAE-Opt-DE).
*Note*: The abbreviations refer to: A - Augmentation, D2A - Dynamic Data Augmentation, BS - Beam Search, H - Heuristics, MCTS - Monte Carlo Tree Search, DE - Differential Evolution, DP - Dynamic Programming, GLS - Guided Local Search, AS - Active Search, AS+M+M - AS+MCTS+Meta-Learning.

similar to theirs. As for **L2C solvers**, NeuOpt (D2A=5, T=3k) beats all baselines on TSP-100 with less time, including the state-of-the-art POMO+EAS+SGBS solver. When pitted against **L2S solvers** which ours also belongs to, NeuOpt is able to at least halve their gaps using a much shorter run time, demonstrating a substantial improvement. Lastly, our NeuOpt (D2A=5, T=5k) is one of the neural solvers (including DPDP) that can solve TSP-100 to near-optimal (i.e., 0.00%) in a reasonable time.

**CVRP results.** Most **L2P solvers** fail to handle CVRP. The remaining ones, CVAE-Opt-DE and DPDP, are significantly outstripped by our solver (D2A=5, T=6k) with less run time. For **L2C solvers**, our NeuOpt-GIRE (D2A=1, T=10k) could exceed the previous state-of-the-art POMO solver. Furthermore, with (D2A=5, T=6k) and (D2A=5, T=40k), our method surpasses POMO+EAS and POMO+EAS+SGBS (long) with less run time, respectively, even though they equip with extra post-hoc per-instance processing boosters. We note that such post-processing strategies (e.g., per-instance active search) could potentially be integrated into ours for further performance gains. Compared to **L2S solvers**, due to the advances of our GIRE, even NeuOpt-GIRE (D2A=1, T=5k) could reduce the gaps of those (masking-based) L2S solvers by an order of magnitude in much less time on CVRP-100. Notably, we are the first **L2S solver** to surpass the strong LKH-3 solver on CVRP.

Table 2: Comparisons of our proposed RDS decoder and GIRE scheme versus existing designs.

| Methods | Encoder Type | Decoder Type | Feasibility Scheme | TSP-100 | | | | CVRP-20 | | | |
|---|---|---|---|---|---|---|---|---|---|---|---|
| | | | | Size(m) | Obj.↓ | Gap↓ | Time↓ | Size(m) | Obj.↓ | Gap↓ | Time↓ |
| DACT (2-opt) | d=64,FF,DAC-Att | DACT | masking | 0.281 | 7.933 | 1.73% | 134s | 0.281 | 6.172 | 0.14% | 58s |
| DACT-U (2-opt) | d=128,MLP,Synth-Att | DACT | masking | 0.633 | 7.822 | 0.32% | 171s | 0.633 | 6.171 | 0.13% | 61s |
| DACT-U-GIRE (2-opt) | d=128,MLP,Synth-Att | DACT | GIRE | 0.633 | 7.822 | 0.32% | 171s | 0.633 | 6.168 | 0.08% | 59s |
| NeuOpt ($K=2$) | d=128,MLP,Synth-Att | RDS | masking | 0.683 | 7.817 | 0.24% | 122s | 0.683 | 6.180 | 0.27% | 60s |
| NeuOpt-GIRE ($K=2$) | d=128,MLP,Synth-Att | RDS | GIRE | 0.683 | 7.817 | 0.24% | 122s | 0.685 | 6.167 | 0.05% | 53s |
| NeuOpt-GIRE ($K=4$) | d=128,MLP,Synth-Att | RDS | GIRE | 0.683 | 7.798 | 0.00% | 133s | 0.685 | 6.163 | 0.00% | 60s |

Table 3: Effects of GRUs, move stream $\mu$, and edge stream $\lambda$.

| Methods | TSP-100 | | CVRP-20 | |
|---|---|---|---|---|
| | Size(M) | Obj.↓ | Size(M) | Obj.↓ |
| w/o-GRUs | 0.468 | 7.804 | 0.470 | 6.165 |
| w/o-$\mu$ | 0.617 | 7.806 | 0.620 | 6.165 |
| w/o-$\lambda$ | 0.617 | 7.799 | 0.620 | 6.164 |
| Ours | 0.683 | 7.798 | 0.685 | 6.163 |

Table 4: Effects of dynamic data augmentation (D2A).

| Inference Type | TSP-100 Gap↓ | CVRP-100 Gap↓ |
|---|---|---|
| w/o-D2A (T=5k) | 0.09% | 1.00% |
| w-D2A (T=5k) | 0.05% | 0.87% |
| w/o-D2A (T=10k) | 0.04% | 0.71% |
| w-D2A (T=10k) | 0.02% | 0.60% |

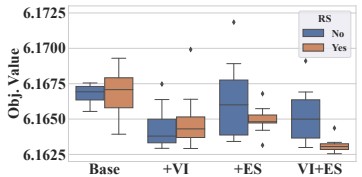

Figure 4: Effects of GIRE designs.

**Effectiveness analysis.** To further reveal the effectiveness of our RDS decoder and GIRE, we conduct a comparison with the state-of-the-art L2S solver, DACT [9]. We upgraded it to DACT-U for fairness, which uses identical embeddings and encoders as our NeuOpt. The evaluation criteria include the model size, the best objective value (averaged over four training runs with T=1k inference steps on 1k validation instances), the gaps w.r.t. the best solver, and the run time. As evidenced in Table 2, our GIRE scheme consistently boosts both NeuOpt and DACT-U on CVRP-20, reducing the mask computation time while marginally increasing model size, which reveals the generality of our GIRE. Moreover, when all are used for 2-opt, NeuOpt ($K = 2$, for TSP) and NeuOpt-GIRE ($K = 2$, for CVRP) surpass DACT-U (for TSP) and DACT-U-GIRE (for CVRP), respectively, with less run time. This is further amplified by increasing $K$ to 4 for NeuOpt at the cost of slightly increased run time.

## 6.2 Ablation studies

**Ablation on RDS decoder.** In Table 3, we remove the key components of our RDS decoder including the GRUs, the move stream $\mu$, and edge stream $\lambda$, respectively, and report the model size and the best objective value averaged over four training runs. As depicted, all components largely contribute to effective contextual modeling for parameterizing the k-opt decoder on both TSP-100 and CVRP-20.

**Ablation on D2A inference.** In Table 4, we gather the gaps (similar to Table 1) with and without our D2A design when a single data augmentation is leveraged during inference with T=5K and T=10K. The results confirm that our D2A consistently enhances the inference on both TSP-100 and CVRP-100.

**Ablation on GIRE designs.** We examine scenarios, considering the inclusion or exclusion of Violation Indicator (VI), Exploration Statistics (ES), and Reward Shaping (RS), using eight combinations of them during the training of NeuOpt on CVRP-20. We draw the box plots of the best objective values achieved across eight training runs for each scenario in Figure 4. We can conclude that: 1) the supplement of VI consistently enhances training outcomes, irrespective of the presence of RS; 2) the supplement of ES may bring improvement only when RS is present; and 3) our complete GIRE design (VI+ES+RS) yields the lowest objective values and demonstrates superior stability.

**Ablation on basis moves.** In Figure 5 (a), we depict the training curves of NeuOpt on TSP-100 with and without the learnable S-move. When removing the learnable S-move and opting for

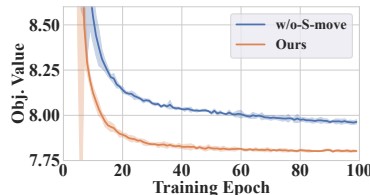

(a) Effects of S-move.

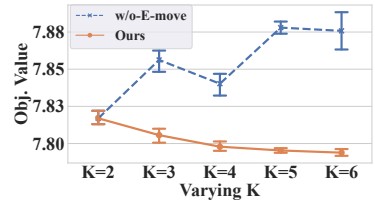

(b) Effects of E-move.

Figure 5: Effects of basis moves.

a random selection of the anchor node $x_a$, a significant degradation in the final performance can be observed, underscoring the necessity of the learnable S-move design. Meanwhile, Figure 5 (b)

Table 5: Generalization (10 runs) on TSPLIB.

| AM -mix | POMO -mix | AMDKD (POMO) | DACT (sol.10k) | | Ours (sol.10k) | |
|---|---|---|---|---|---|---|
| | | | Avg.↓ | Best↓ | Avg.↓ | Best↓ |
| 19.59% | 0.92% | 1.18% | 3.29% | 1.59% | 0.85% | 0.50% |

Table 6: Generalization (10 runs) on CVRPLIB.

| AM -mix | POMO -mix | AMDKD (POMO) | DACT (sol.10k) | | Ours (sol.10k) | |
|---|---|---|---|---|---|---|
| | | | Avg.↓ | Best↓ | Avg.↓ | Best↓ |
| 15.87% | 8.05% | 5.77% | 5.21% | 3.68% | 4.80% | 3.27% |

Table 7: Results on $N = 200$.

| Methods | TSP-200 | | CVRP-200 | |
|---|---|---|---|---|
| | Gap.↓ | Time.↓ | Gap.↓ | Time.↓ |
| LKH [20, 51] | 0.00% | 2.3h | 1.17% | 21.6h |
| Ours (D2A=5,T=10k) | 0.04% | 4.7h | 0.68% | 9.6h |
| Ours (D2A=5,T=20k) | 0.02% | 9.4h | 0.48% | 19.2h |
| Ours (D2A=5,T=30k) | 0.01% | 14.1h | 0.39% | 1.2d |

Table 8: Influence of $K$.

| $K$ | T(inf.)↓ | T(tr.)↓ | Gap↓ |
|---|---|---|---|
| 2 | 2m02s | 20m | 0.30% |
| 3 | 2m07s | 21m | 0.15% |
| 4 | 2m13s | 22m | 0.05% |
| 5 | 2m19s | 24m | 0.01% |
| 6 | 2m26s | 25m | 0.00% |

Figure 6: Influence of $\alpha$ and $\beta$.

presents the pointplots with confidence intervals that show the performance of our NeuOpt on TSP-100 with and without E-move during decoding across varying preset $K$. When E-move is absent (dotted blue line), the model, downgraded to performing fixed $K$-opt only, exhibits diminished performance on larger $K$. Conversely, our NeuOpt (solid orange line) could further benefit from a larger $K$, due to its flexibility in determining and combining different $k$ across search steps.

## 6.3 Generalization and scalability studies

In Table 5 and Table 6, we further evaluate the generalization capability of our NeuOpt models on more complex instances from TSPLIB [55] and CVRPLIB [56], respectively. Our NeuOpt achieves lower gaps than DACT [9] and L2C solvers (AM [3] and POMO [4]), and even shows superiority over the AMDKD method [11] that is explicitly designed to boost the generalization of POMO through knowledge distillation. Beyond generalization, our NeuOpt also exhibits notable scalability. As shown in Table 7, when trained directly for size 200, NeuOpt finds close-to-optimal solutions and still surpasses the strong LKH-3 solver [20] on CVRP-200. Note that existing L2S solvers, e.g., DACT [9] may struggle with training on such scales. More details and discussions are available in Appendix E.

## 6.4 Hyper-parameter studies

**Influence of preset $K$.** In Table 8, we display the performance of NeuOpt on TSP-100, as well as the corresponding inference time (T=1k) and the training time (per epoch) for varying $K$ values. The results highlight trade-offs between better performance and increased computational costs.

**Influence of GIRE hyper-parameters.** Figure 6 depicts the influence of $\alpha$ and $\beta$ in Eq. (4) on CVRP-20, where we fix one while varying the other, investigating both extremely smaller (0.01) and larger (0.1) values. The results suggest that more effective reward shaping occurs when the weights are moderate. Please refer to Appendix D for discussions on more GIRE hyper-parameters.

## 7 Conclusions and limitations

In this paper, we introduce NeuOpt, a novel L2S solver for VRPs, that performs flexible k-opt exchanges with a tailored formulation and a designed RDS decoder. We also present GIRE, the first scheme to transcend masking for constraint handling based on feature supplement and reward shaping, enabling autonomous exploration in both feasible and infeasible regions. Moreover, we devise a D2A augmentation method to boost inference diversity. Despite delivering state-of-the-art results, our work still has **limitations**. While NeuOpt exhibits better scalability than existing L2S solvers, it falls short against some L2P solvers (e.g., [6, 7]) for larger-scale TSPs. Possible solutions in future works include: 1) integrating divide-and-conquer strategies as per [6, 57–60], 2) reducing search space via heatmaps as predicted in [6, 15], 3) adopting more scalable encoders [61, 62], and 4) refactoring our code with highly-optimized CUDA libraries as did in [7, 15]. Besides enhancing scalability, future works can also focus on: 1) applying our GIRE to more VRP constraints and even beyond L2S solvers, 2) integrating our method with post-hoc per-instance processing boosters (e.g., EAS [5]) for better performance, and 3) enhancing the generalization capability of our NeuOpt on instances with different sizes/distributions (e.g., by leveraging the frameworks in [11, 63]).

## Acknowledgments and Disclosure of Funding

This research was supported by the Singapore Ministry of Education (MOE) Academic Research Fund (AcRF) Tier 1 grant.

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

# Learning to Search Feasible and Infeasible Regions of Routing Problems with Flexible Neural k-Opt (Appendix)

## A  Action factorization examples

Figure 7 depicts examples of our factorization method using combinations of basis moves (S-move, I-move, and E-move) to represent different k-opt exchanges. Initiated from a TSP-9 instance (leftmost), we list examples from 1-opt to 4-opt, where the number of I-move corresponds to varying $k$ values, leading to distinct new solutions (rightmost). This demonstrates the flexibility of our factorization.

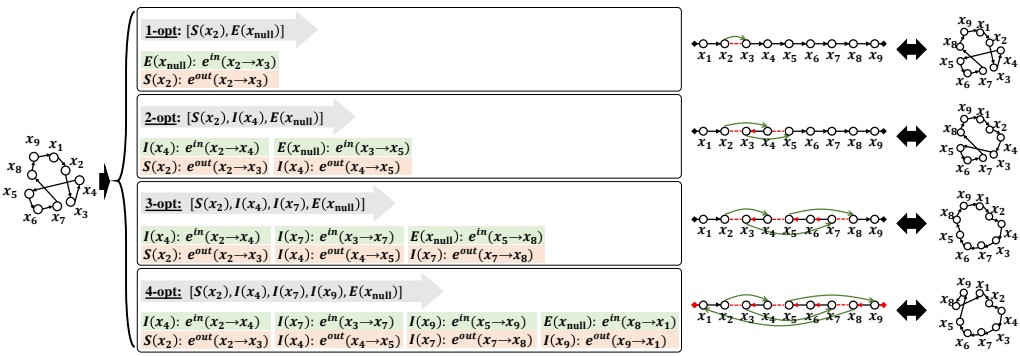

Figure 7: Examples of using the basis moves to factorize 1-opt (void action), 2-opt, 3-opt, and 4-opt.

## B  NeuOpt encoder

Given a state $s_t = \{\mathcal{G}, \tau_t, \tau_t^{\text{bsf}}\}$, where $\mathcal{G}$ is the current instance, $\tau_t$ is the current solution, and $\tau_t^{\text{bsf}}$ is the best-so-far solution before step $t$, the encoding process first translates the raw features of state into node embeddings. These embeddings are subsequently refined through $L = 3$ stacked encoders.

**Fearure Embedding.** Building upon the dual-aspect representation design from [12], we embed state features[3], denoted as $\psi[\mathcal{G}, \tau_t] = \{\{\varphi_i^t\}_{x_i \in \mathcal{V}}, \{p_i^t\}_{x_i \in \mathcal{V}}\}$ into two separate sets of node embeddings: *Node Feature Embeddings* (NFEs) to encode $\psi(\mathcal{G}) = \{\varphi_i^t\}$ and *Positional Feature Embeddings* (PFEs) to encode $\psi(\tau_t) = \{p_i^t\}$. The **NFEs**, denoted as $\{h_i^{\text{init}}\}_{x_i \in \mathcal{V}}$, are a set of $d$-dimensional vectors, each of which embeds $d_h$-dimensional problem-specific raw features $\varphi_i^t$ of node $x_i$ at step $t$. To obtain $h_i^{\text{init}}$, we upgrade linear projection used in [12] into the MLP with structure of $(d_h \times \frac{d}{2} \times d)$ for enhanced representation. The **PFEs**, denoted as $\{g_i^{\text{init}}\}_{x_i \in \mathcal{V}}$, are a set of $d$-dimensional vectors, each of which embeds positional features $p_i^t$ of $x_i$ at step $t$, derived from the position of $x_i$ in the current solution $\tau_t$. Following [9], we employ the Cyclic Positional Encoding (CPE) to generate a series of cyclic embeddings in a $d$-dimensional space. These CPE embeddings are then used to initialize $g_i^{\text{init}}$ correspondingly, so as to capture the topological structure of the nodes in the current solution $\tau_t$.

**Problem-specific raw node features $\varphi_i^t$.** For **TSP**, node features $\varphi_i^t$ of $x_i$ contains its two-dimensional coordinates (i.e., $d_h = 2$); For **CVRP**, $\varphi_i^t$ contains six features (i.e., $d_h = 6$) including 1-2) its two-dimensional coordinates, 3) the demand of node $x_i$, 4) the sum of demand of the corresponding sub-tour before node $x_i$ (inclusive), 5) the sum of demand of the corresponding sub-tour after $x_i$ (exclusive), and 6) an indicator function to signify whether node $x_i$ is a customer node. When GIRE is applied, we enrich $\varphi_i^t$ with two Violation Indicator (VI) features, i.e., $d_h = 8$.

---

[3]Note that the $\tau_t^{\text{bsf}}$ is leveraged in the reward function and the critic network, but not the policy network.

**Stacked encoders.** Following the encoders in [9, 12], we use Transformer-styled encoders with Synthesis Attention (Synth-Att) to refine the embeddings $\{h_i^{\text{init}}\}x_i \in \mathcal{V}$ and $\{g_i^{\text{init}}\}x_i \in \mathcal{V}$, where PFEs serve as auxiliary embeddings that bolster the representation learning of NFEs. After the encoding, a unified set of embeddings $\{h_i\}_{x_i \in \mathcal{V}}$ is obtained, which is then inputted in our recurrent dual-stream (RDS) decoder for k-opt action decoding (see Section 4).

## C  Training and inference algorithms

### C.1  Training algorithm

---

**Algorithm 1** Reinforcement learning algorithms for NeuOpt

---

**Input**: policy network $\pi_\theta$, critic network $v_\phi$, PPO objective clipping threshold $\vartheta$, learning rate $\eta_\theta, \eta_\phi$, learning rate decay rate $\varsigma$, number of PPO inner loops $\Omega$, training steps $T_{\text{train}}$, number of epochs $E$, number of batches per epoch $B$, curriculum learning (CL) scalar $\xi$

1: **for** $epoch = 1$ to $E$ **do**
2:   **for** $batch = 1$ to $B$ **do**
3:     Randomly generate a batch of training instances $\mathcal{D} = \{\mathcal{G}_i\}$ and their initial solutions $\{\tau_i\}$;
4:     **CL**: Improve $\{\tau_i\}$ to $\{\tau_i'\}$ by runing the current policy $\pi_\theta$ for $T = epoch/\xi$ steps;
5:     Get initial state $s_0$ based on $\{\tau_i'\}$ for each instance in the current batch;
6:     $t \leftarrow 0$;
7:     **while** $t < T_{\text{train}}$ **do**
8:       Run policy $\pi_\theta$ on each instance and get $\{(s_{t'}, a_{t'}, r_{t'})\}_{t'=t}^{t+n-1}$ where $a_{t'} \sim \pi_\theta(a_{t'}|s_{t'})$;
9:       $t \leftarrow t + n, \pi_{old} \leftarrow \pi_\theta, v_{old} \leftarrow v_\phi$;
10:       **for** $j = 1$ to $\Omega$ **do**
11:         $\hat{R}_t = v_\phi(s_t)$;
12:         **for** $t' \in \{t-1, ..., t-n\}$ **do**
13:           $\hat{R}_{t'} \leftarrow r_{t'} + \gamma \hat{R}_{t'+1}$;
14:           $\hat{A}_{t'} \leftarrow \hat{R}_{t'} - v_\phi(s_{t'})$;
15:         **end for**
16:         Compute RL loss $\mathcal{L}_\theta^{\text{PPO}}$ using Eq. (6) and critic loss $\mathcal{L}_\phi^{\text{Critic}}$ using Eq. (7);
17:         $\theta \leftarrow \theta + \eta_\theta \nabla \mathcal{L}_\theta^{\text{PPO}}; \phi \leftarrow \phi - \eta_\phi \nabla \mathcal{L}_\phi^{\text{Critic}}$;
18:       **end for**
19:     **end while**
20:   **end for**
21:   $\eta_\theta \leftarrow \varsigma \eta_\theta, \eta_\phi \leftarrow \varsigma \eta_\phi$;
22: **end for**

---

As detailed in Algorithm 1, we adapt the $n$-step proximal policy optimization (PPO) with curriculum learning (CL) strategy used in [9, 12] to train our NeuOpt. For our GIRE scheme, we consider learning separate critics $v_\phi^{\text{origin}}$, $v_\phi^{\text{reg}}$, and $v_\phi^{\text{bonus}}$ to fit the respective reward shaping terms in Eq.(4), so as to better estimate the state values. In light of this, when GIRE is applied, we update Algorithm 1 by: 1) duplicating lines 11, 13-14 to compute $\hat{R}_{t'}^{\text{origin}}$, $\hat{R}_{t'}^{\text{reg}}$, and $\hat{R}_{t'}^{\text{bonus}}$ as well as $\hat{A}_{t'}^{\text{origin}}$, $\hat{A}_{t'}^{\text{reg}}$, and $\hat{A}_{t'}^{\text{bonus}}$ at the same time; 2) updating the line 16 to replace Eq. (6) and Eq. (7) into Eq. (8) and Eq. (9), respectively; and update line 17 accordingly where all the critics share the same learning rate $\eta_\phi$.

$$\mathcal{L}_\theta^{\text{PPO}} = \frac{1}{n|\mathcal{D}|} \sum_{\mathcal{D}} \sum_{t'=t-n}^{t-1} \min\left( \frac{\pi_\theta(a_{t'}|s_{t'})}{\pi_{old}(a_{t'}|s_{t'})} \hat{A}_{t'}, \text{ Clip}\left[ \frac{\pi_\theta(a_{t'}|s_{t'})}{\pi_{old}(a_{t'}|s_{t'})}, 1-\vartheta, 1+\vartheta \right] \hat{A}_{t'} \right), \quad (6)$$

$$\mathcal{L}_\phi^{\text{Critic}} = \frac{1}{n|\mathcal{D}|} \sum_{\mathcal{D}} \sum_{t'=t-n}^{t-1} \max\left( \left| v_\phi(s_{t'}) - \hat{R}_{t'} \right|^2, \ \left| v_\phi^{\text{Clip}}(s_{t'}) - \hat{R}_{t'} \right|^2 \right). \quad (7)$$

$$\mathcal{L}_\theta^{\text{GIRE}} = \frac{1}{n|\mathcal{D}|} \sum_{\mathcal{D}} \sum_{t'=t-n}^{t-1} \min\left( \frac{\pi_\theta(a_{t'}|s_{t'})}{\pi_{old}(a_{t'}|s_{t'})} \left( \hat{A}_{t'}^{\text{origin}} + \hat{A}_{t'}^{\text{red}} + \hat{A}_{t'}^{\text{bonus}} \right), \right.$$
$$\left. \text{Clip}\left[ \frac{\pi_\theta(a_{t'}|s_{t'})}{\pi_{old}(a_{t'}|s_{t'})}, 1-\vartheta, 1+\vartheta \right] \left( \hat{A}_{t'}^{\text{origin}} + \hat{A}_{t'}^{\text{red}} + \hat{A}_{t'}^{\text{bonus}} \right) \right), \quad (8)$$

$$\mathcal{L}_\phi^{\text{GIRE}} = \frac{1}{n|\mathcal{D}|} \sum_{\mathcal{D}} \sum_{t'=t-n}^{t-1} \max \Big($$

$$\left| v_\phi^{\text{origin}}(s_{t'}) - \hat{R}_{t'}^{\text{origin}} \right|^2 + \left| v_\phi^{\text{reg}}(s_{t'}) - \hat{R}_{t'}^{\text{reg}} \right|^2 + \left| v_\phi^{\text{bonus}}(s_{t'}) - \hat{R}_{t'}^{\text{bonus}} \right|^2, \qquad (9)$$

$$\left| v_\phi^{\text{origin,clip}}(s_{t'}) - \hat{R}_{t'}^{\text{origin}} \right|^2 + \left| v_\phi^{\text{reg,clip}}(s_{t'}) - \hat{R}_{t'}^{\text{reg}} \right|^2 + \left| v_\phi^{\text{bonus,clip}}(s_{t'}) - \hat{R}_{t'}^{\text{bonus}} \right|^2 \Big).$$

### C.2  Inference algorithm

In our previous work [12], the data augmentation was incorporated during inference to boost the search diversity. The rationale is that a specific instance, $\mathcal{G}$, can be transformed to different ones, yet still retain the identical optimal solution. These augmented instances can be solved differently by the trained model in parallel, thereby enhancing the diversity of the search for better performance. Specifically, each augmented instance is generated by consecutively executing four predetermined invariant augmentation transformations as listed in Table 9, where the execution order and configurations used for each transformation are randomly determined on the fly following Algorithm 2.

Table 9: Descriptions of the four invariant augmentation transformations (retrieved from [12]).

| Transformations | Formulations | Configurations |
|---|---|---|
| flip-x-y | $(x', y') = (y, x)$ | perform or skip |
| 1-x | $(x', y') = (1 - x, y)$ | perform or skip |
| 1-y | $(x', y') = (x, 1 - y)$ | perform or skip |
| rotate | $\begin{pmatrix} x' \\ y' \end{pmatrix} = \begin{pmatrix} x \cos\boldsymbol{\theta} - y \sin\boldsymbol{\theta} \\ x \sin\boldsymbol{\theta} + y \cos\boldsymbol{\theta} \end{pmatrix}$ | $\boldsymbol{\theta} \in \{0, \pi/2, \pi, 3\pi/2\}$ |

---

**Algorithm 2** Augmentation (retrieved from [12])

---

**Input**: Instance $\mathcal{G}$
**Output**: Augmented instance $\mathcal{G}'$
1: $\mathcal{G}' \leftarrow \mathcal{G}$;
2: $\mathcal{A} \leftarrow$ **RandomShuffle**([flip-x-y, 1-x, 1-y, rotate]);
3: **for** each augment method $j \in \mathcal{A}$ **do**
4: $\quad \Im(j) \leftarrow$ **RandomConfig**$(j)$;
5: $\quad \mathcal{G}' \leftarrow$ perform augment $j$ on $\mathcal{G}'$ with config $\Im(j)$;
6: **end for**

---

However, such augmentation is performed only once at the start of the inference, making the augmentation fixed across the search process. We thus propose Dynamic Data Augmentation (D2A). It suggests generating new augmented instances with different augmentation configurations each time when the solver fails to find a better solution within a consecutive maximum of $T_{\text{D2A}}$ steps (i.e., we consider the search to be trapped in local optima). This allows the model to explicitly solve instances differently once it gets trapped in the local optima, thus promoting an even more diverse search process. Note that the proposed D2A is generic to boost most L2S solvers, and even has the potential to boost L2C solvers when equipped with post-hoc per-instance processing boosters such as EAS [5]. In Algorothm 3, we summarize the D2A procedures where we note that the loops in line 1 and line 7 can be run in parallel during practical implementations. For example, when we use "D2A=5" as per in Table 1, it means that for each instance $\mathcal{G}$, there are 5 augmented instances $\{\mathcal{G}_1, \mathcal{G}_2, \mathcal{G}_3, \mathcal{G}_4, \mathcal{G}_5\}$ being solved simultaneously. Each of these instances has its own counter $T_i^{\text{stall}}$ and the best-so-far solution $\tau_i^{\text{bsf}}$ to track whether the search (for the particular instance $\mathcal{G}_i$) has fallen into local optima.

## D  Additional discussions on GIRE scheme

**Separate critic networks.** As per Appendix C, we use separate critics in GIRE. We now introduce their detailed architectures. During the critic value estimation, we first upgrade embeddings $\{h_i\}$

**Algorithm 3** Dynamic Data Augmentation (D2A)

---

**Input**: Instance $\mathcal{G}$, policy network $\pi_\theta$, inference step $T$, number of augments $D2A$, maximum number of consecutive steps allowed before considering the search trapped in local optima $T_{\text{D2A}}$
**Output**: Best solution found during solving all the augmented instances $\mathcal{G}_i$

1: **for** $i = 1, \cdots, D2A$ **do**
2:     Get an augmented instance: $\mathcal{G}_i \leftarrow$ **Augmentation**$(\mathcal{G})$;
3:     Get a random solution $\tau_{i,0}$ and set it as the best-so-far solution for $\mathcal{G}_i$: $\tau_i^{\text{bsf}} \leftarrow \tau_{i,0}$;
4:     Set counter: $T_i^{\text{stall}} \leftarrow 0$;
5: **end for**
6: **for** $t = 1, \cdots, T$ **do**
7:     **for** $i = 1, \cdots, D2A$ **do**
8:         Run one inference step to get a new solution $\tau_{i,t}$ for $\mathcal{G}_i$ using policy network $\pi_\theta$;
9:         **if** new solution $\tau_{i,t}$ is a new best-so-far solution for instance $\mathcal{G}_i$ **then**
10:             Update the best-so-far solution: $\tau_i^{\text{bsf}} \leftarrow \tau_{i,t}$;
11:             Reset counter: $T_i^{\text{stall}} \leftarrow 0$;
12:         **else**
13:             Increment counter: $T_i^{\text{stall}} = T_i^{\text{stall}} + 1$;
14:         **end if**
15:         **if** $T_i^{\text{stall}} \geq T_{\text{D2A}}$ **then**
16:             Get a new augmented instance: $\mathcal{G}_i \leftarrow$ **Augmentation**$(\mathcal{G})$;
17:             Reset counter $T_i^{\text{stall}} \leftarrow 0$
18:         **end if**
19:     **end for**
20: **end for**

---

(from the encoders of $\pi_\theta$) to $\{\hat{h}_i\}_{x_i \in \mathcal{V}}$ using a vanilla multi-head attention layer. The $\{\hat{h}_i\}$ are then fed into a mean-pooling layer [39], yielding compressed representations as follows,

$$\hat{y}_i = \hat{h}_i W^{Local} + \text{mean}\left[\{\hat{h}_i\}_{x_i \in \mathcal{V}}\right] W^{Global}, \tag{10}$$

where $W \in \mathbb{R}^{d \times \frac{d}{2}}$ are trainable matrices, and the mean and max are element-wise operators. All critics $v_\phi^{\text{origin}}$, $v_\phi^{\text{reg}}$, and $v_\phi^{\text{bonus}}$ then share these representations $\{\hat{y}_i\}$ as parts of their inputs. Specifically, $v_\phi^{\text{origin}}$ is computed via a four-layer MLP in Eq. (11) with structure $(d + 1, d, \frac{d}{2}, 1)$ which leverages $f(\tau_t^{\text{bsf}})$ as additional features; $v_\phi^{\text{reg}}$ is computed via a four-layer MLP in Eq. (12) with structure $(d+10, d, \frac{d}{2}, 1)$ which leverages $f(\tau_t^{\text{bsf}})$ and ES features $\mathcal{J}_t$ as additional features; and $v_\phi^{\text{bonus}}$ is computed via a four-layer MLP in Eq. (13) with structure $(d + 1, d, \frac{d}{2}, 1)$ which leverages the best-so-far cost w.r.t. $\epsilon$-$\mathcal{F}$ regions $f(\tau_t^{\text{bsf-wrt-}\epsilon})$ as additional features[4]. All MLPs use the ReLU activation function.

$$v_\phi^{\text{origin}} = \text{MLP}_{\phi_1}\left(\max[\{\hat{y}_i\}_{x_i \in \mathcal{V}}], \text{mean}[\{\hat{y}_i\}_{x_i \in \mathcal{V}}], f(\tau_t^{\text{bsf}})\right) \tag{11}$$

$$v_\phi^{\text{reg}} = \text{MLP}_{\phi_2}\left(\max[\{\hat{y}_i\}_{x_i \in \mathcal{V}}], \text{mean}[\{\hat{y}_i\}_{x_i \in \mathcal{V}}], f(\tau_t^{\text{bsf}}), \mathcal{J}_t\right) \tag{12}$$

$$v_\phi^{\text{bonus}} = \text{MLP}_{\phi_3}\left(\max[\{\hat{y}_i\}_{x_i \in \mathcal{V}}], \text{mean}[\{\hat{y}_i\}_{x_i \in \mathcal{V}}], f(\tau_t^{\text{bsf-wrt-}\epsilon})\right) \tag{13}$$

**Illustrations of extreme search behaviour and our GIRE efficacy.** Recall that our GIRE considers reward-shaping terms to encourage search within more promising feasibility boundaries and regulate extreme search behaviours using an entropy measure of the estimated conditional transforming probabilities $P_t(\mathcal{U}|\mathcal{U}) = P(\tau' \in \mathcal{U}|\tau \in \mathcal{U})$ and $P_t(\mathcal{F}|\mathcal{F}) = P(\tau' \in \mathcal{F}|\tau \in \mathcal{F})$. Figure 8 depicts the persistence of extreme search behaviour when RS is absent while validating the efficacy of our reward shaping (RS) in GIRE. We conduct training with and without RS, recording the convergence curves of the objective values (with mean and confidence intervals) in Figure 8(a), the detailed objective values per run in Figure 8(b), the estimated probability $P_t(\mathcal{F}|\mathcal{F})$ in Figure 8(c), and the estimated probability $P_t(\mathcal{U}|\mathcal{U})$ in Figure 8(d). As revealed in Figure 8(a), without RS, the runs show unstable convergence and poorer final objective values. This can be attributed to higher $P_t(\mathcal{F}|\mathcal{F})$ and lower $P_t(\mathcal{U}|\mathcal{U})$ as shown in Figure 8(c) and Figure 8(d), indicating biased search preference towards feasible regions

---

[4]Given that GIRE uses additional features, the MDP states should be augmented accordingly.

and inefficient exploration in infeasible regions. In the worst case, training may fail to converge to lower objective values due to extremely low $P_t(\mathcal{U}|\mathcal{U})$ (below 0.1 as shown in Figure 8(d)), indicating extreme search behaviour and entrapment in local optima. Conversely, GIRE fosters moderate search behaviours and promotes exploration in infeasible regions (especially for the boundaries), leading to significantly improved training curves with reduced variance and lower objective values.

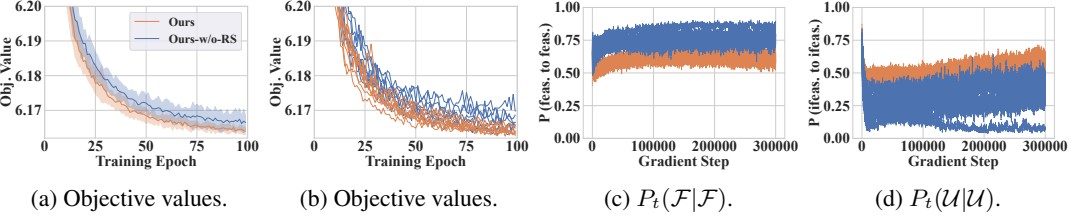

(a) Objective values.  (b) Objective values.  (c) $P_t(\mathcal{F}|\mathcal{F})$.  (d) $P_t(\mathcal{U}|\mathcal{U})$.

Figure 8: Training curves of our GIRE with and without RS on CVRP-20 (8 runs).

**Influence of $\epsilon$.** We employ a deterministic rule to decide $\epsilon$, i.e., $\epsilon = \zeta \times N_{\text{customer}} \times (\bar{\delta}/\Delta)$, where $\zeta$ is a coefficient within $[0, 1]$, $\Delta$ is the vehicle capacity, and $\bar{\delta}$ represents the average customer demands. This formula implies that the total violation corresponds to scenarios where $\zeta$-ratio of customers, each with the average demand, breaches the constraint. Empirically, we set $\zeta = 0.1$. Figure 9 illustrates the impact of varying $\zeta$.

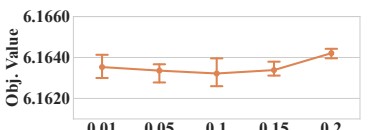

Figure 9: Impact of varying $\zeta$.

**Influence of $c_1$ and $c_2$.** Recall that in the entropy measure defined by Eq. (5), we introduce two hyper-parameters $c_1$ and $c_2$ to shape the entropy measure pattern $\mathbb{H}[P]$, where the measure $\mathbb{H}[P]$ is used to penalize extreme search behaviours, particularly when values of $P$ approach 0 or 1. Figure 10 illustrates how hyper-parameters $c_1$ and $c_2$ influence the shape of $\mathbb{H}[P]$. When $c_2$ is fixed, a smaller $c_1$ expands the penalty range, while a larger $c_1$ constricts the penalty range. Similarly, when $c_1$ is fixed, a smaller or a larger $c_2$ will also control the shape of the patterns. Empirically, we set the values of $c_1 = 0.5, c_2 = 2.5$ so as to only penalize extreme search behaviour if the feasibility transition probability is outside the $[0.25, 0.75]$ range. In Figure 11, we investigate the influence of $c_1, c_2$ on the training stability of our NeuOpt-GIRE approach on CVRP-20. Results show that they may not affect training stability (thus no need for extensive tuning in practical usage).

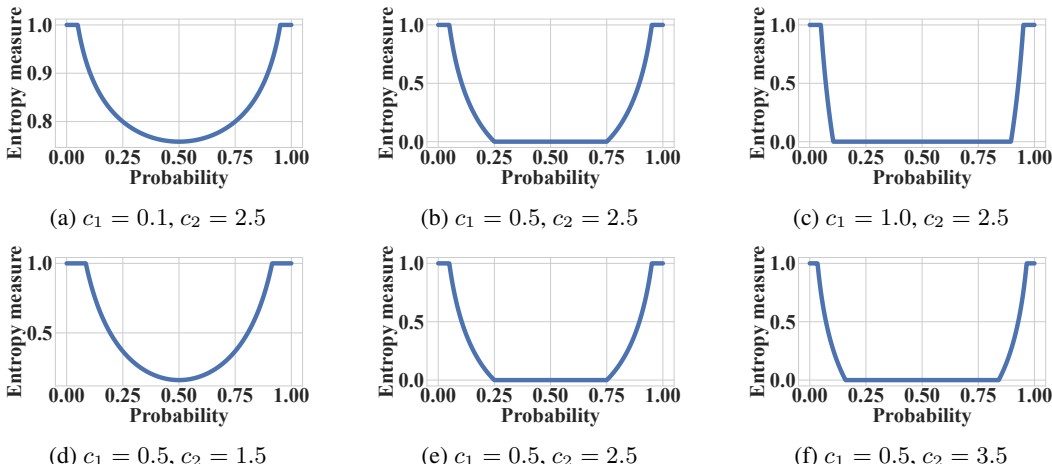

(a) $c_1 = 0.1, c_2 = 2.5$  (b) $c_1 = 0.5, c_2 = 2.5$  (c) $c_1 = 1.0, c_2 = 2.5$

(d) $c_1 = 0.5, c_2 = 1.5$  (e) $c_1 = 0.5, c_2 = 2.5$  (f) $c_1 = 0.5, c_2 = 3.5$

Figure 10: Effects of $c_1$ and $c_2$ on $\mathbb{H}[P]$ pattern: (a)-(c) fix $c_2 = 2.5$ and vary $c_1$; (d)-(f) fix $c_1 = 0.5$ and vary $c_2$. Compared to the pattern (b) and (e) used in this paper, varying $c_1$ and $c_2$ either **constricts** the penalty range, shown in (c) and (f), or **expands** the penalty range, shown in (a) and (d).

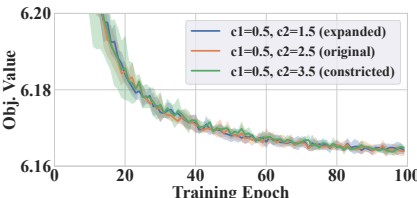

Figure 11: Influence of **constricted** and **expanded** patterns of $\mathbb{H}[P]$ on the training stability.

**Applying GIRE to other constraints.** Our GIRE can be viewed as an important early attempt that moves beyond the pure feasibility masking to autonomously explore both feasible and infeasible regions during the search. To adapt GIRE for a particular constraint, we suggest the following:

- For **Feature Supplement**, simply binary indicator functions can be employed to discern specific constraint violations of each node in the solution, thereby forming the Violation Indicator (VI) features. For example, VI can indicate nodes (customers) in the solution that breach their time window or the pickup/delivery precedence constraints. For the Exploration Statistics (ES) features, they can be retained as originally conceptualized since they are based on historical exploration records, rather than specific constraints.

- For **Reward Shaping**, it can be retained as originally conceptualized. We suggest characterizing the $\epsilon$-feasible regions by a fraction of nodes that do not adhere to the constraints, i.e., $\zeta = 0.1$.

## E  Additional experimental results

### E.1  Details of implementation

**Hyper-parameter details.** We use $d = 128$, $T_{\text{D2A}} = 10$, $T_{\text{his}} = 25$ for NeuOpt, and employ 4 attention heads in both the encoders and critics. The ReLU activation function is utilized for MLPs within the network. In line with the training algorithms in [9, 12], we retain their hyper-parameters to ensure an equivalent training amount. Training involves $E = 200$ epochs and $B = 20$ batches per epoch, with batch sizes of 512 (TSP) and 600 (CVRP). For TSP, we use $n = 4$, $T_{\text{train}} = 200$; and for CVRP, we use $n = 5$, $T_{\text{train}} = 250$. The PPO inner loops is $\Omega = 3$ and the clip threshold is $\vartheta = 0.1$. The Adam optimizer is employed with learning rates $\eta_\theta = 8 \times 10^{-5}$ for $\pi_\theta$, $\eta_\phi = 2 \times 10^{-5}$ for $v_\phi$, as well as a decay rate $\varsigma = 0.985$. The reward discount factor $\gamma$ is set to 0.999 for both problems. Besides, we empirically set approach-specific parameters: the gradient norm of NeuOpt is clipped at 0.05 in all cases, and the curriculum learning scalar $\xi$ is set as 1, 0.5, and 0.25 for sizes 20, 50, and 100, respectively. The training time varies depending on the specific problem and size, e.g., CVRP requires about 4 days for size 20 (1 GPU), 5 days for size 50 (2 GPUs), and 8 days for size 100 (4 GPUs), which are around half the time taken as reported in DACT [9] and POMO [4] on CVRP100.

**Setup details.** We follow Kool et al. [3] to sample all training and test instance coordinates within the unit square $[0, 1] \times [0, 1]$ uniformly. For CVRP, the demands of customers are uniformly sampled from the set $\{1, 2, ..., 9\}$ and the vehicle capacity is set to 30, 40, and 50 for sizes 20, 50, and 100 respectively (we thus estimate $\bar{\delta}$ as 5/30, 5/40, 5/50 in our GIRE, respectively). To facilitate GPU parallelization given the varying lengths of CVRP solutions, we employ the *dummy depots* design in [9, 39] for length padding, where depots are duplicated during both training and inference. Same as DACT [9], we include 10, 20, and 20 dummy depots for CVRP sizes of 20, 50, and 100 respectively.

**Baseline details.** We provide details on the compared neural baselines. For the ones with #, we found certain **issues** in their code, which may lead to the underestimation of reported gaps by around 0.02% (TSP-50) and 0.04% (TSP-100). We observed that they used 'GEO' (Geographical distance) instead of 'EUC' (Euclidean distance) as the type of edge weights while running the Concorde to determine the optimal solutions. This results in the optimal solutions (optimal in the 'GEO' setting) no longer being optimal in the 'EUC' settings, while the objectives derived by neural solvers are based on Euclidean distances. This **discrepancy** might be the reason for the reported **negative** optimality gaps on TSP, which seems anomalous as Concorde is guaranteed to produce optimal solutions.

- **GCN+BS** [14][#]: a classic L2P solver that predicts heatmaps for TSP based on GCN. We run its star version that considers both beam search (1,280 beams) and shortest tour heuristics.

- **Att-GCN+MCTS** [6]#: an efficient L2P solver that generalizes heatmap-based L2P solvers to larger-scale TSP instances through divide-and-conquer. However, we encountered issues running their GPU-version code, and the CPU-version code appeared to be time-consuming, taking around 28h (10s/instance) for 10k TSP100. We thus opt to use their reported results directly.
- **GNN+GLS** [40]: an insightful L2P solver that uses GNN to predict regrets, thereby directing the traditional local search (relocate and 2-opt) heuristics for TSP. Regrettably, the code is not optimized for batch inference and we encountered some issues loading our datasets when following their instructions. As such, we opt to use their reported results directly.
- **CVAE-Opt-DE** [43]: a generic L2P solver that learns to predict a latent search space for TSP and CVRP. We use their reported DE version results due to the long runtime to reproduce results.
- **DPDP** [42]: a state-of-the-art L2P solver that combines heatmaps with dynamic programming to efficiently solve TSP and CVRP. We run it for 100k iterations (TSP) and 1,000k iterations (CVRP).
- **DIMES** [7]#: a novel L2P solver that learns to predict a latent search space and exploits differentiable optimizers to find TSP solutions. Due to incompatible issues when installing the required Pytorch extension library packages, we opt to directly use their reported results obtained by the settings of using REINFORCE, active search, Monte Carlo tree search, and meta-learning.
- **DIFUSCO** [15]#: a state-of-the-art L2P solver, that learns a diffusion model to predict heatmaps for TSP. We run their used setting: T=50 diffusion steps, S=16 samplings, and 2-opt post-processing.
- **AM+LCP**★ [33]: a two-stage L2C solver that iteratively constructs (seeder) and re-constructs (reviser) parts of the solutions based on the AM [3]. For TSP, we run their optimized version with 1,280 samplings and two revisers - one revises a length of 10 nodes for 25 iterations, and the other revises a length of 20 nodes for 20 iterations. However, their code for CVRP is not publicly available. Hence for CVRP, we opt to directly use their reported results.
- **Pointformer** [32]: a latest L2C solver that enhances POMO with a multi-pointer Transformer and feature augmentation for TSP instances. We run it with 200 samplings and $\times 8$ augmentations.
- **Sym-NCO** [13]: an effective L2C solver that enhances POMO with auxiliary losses for TSP and CVRP, aiming at better symmetry handling. We run it with 200 samplings and $\times 8$ augmentations.
- **POMO** [4]: a renowned L2C solver that enhances the AM [3] with diverse rollouts and data augmentations for TSP and CVRP. We run it with 200 samplings and $\times 8$ augmentations.
- **POMO+EAS** [5]: a leading L2C solver that adjusts a small subset of the pre-trained POMO model parameters on test instances for efficient active search. We run the EAS-lay version with $\times 8$ augmentations for T=200 iterations (both TSP and CVRP) which is the best setting for CVRP.
- **POMO+EAS+SGBS** [34]: a state-of-the-art L2C solver that bolsters EAS by incorporating simulation-guided beam search, executing beam search and efficient active search phases alternately. We run 5 rounds (TSP) and 50 rounds (CVRP) for the short version, and 20 rounds (TSP) and 200 rounds (CVRP) for the long version. The beam search width is set as per their suggested values.
- **Costa et al.** [16]: a compelling L2S solver that employs RNN-based policy to control the 2-opt for solving TSP. We run it for T=2k iterations.
- **Sui et al.** [17]: a recent L2S solver improves upon [16] by performing 3-opt, where the policy first removes three edges and then chooses a re-connection type from all candidate options. Given that their code is not available, we opt to directly use their reported results.
- **Wu et al.** [39]: a classic L2S solver that employs Transformer-based policy to control the 2-opt for solving TSP. We run it for T=5k iterations (both TSP and CVRP).
- **DACT** [9]: a state-of-the-art L2S solver that enhances Wu et al. [39] via the cyclic positional encoding and dual-aspect representations. Consistent with their original best settings, we run it for 10k iterations with $\times 4$ augmentations for TSP and $\times 6$ augmentations for CVRP.
- **NLNS** [8]: a strong L2S solver that learns to control ruin-and-repair operators. Following their default settings, we run their code on 10 CPUs in parallel, executing T=5k iterations.
- **NCE** [37]: a distinctive L2S solver that is designed to execute CROSSOVER exchanges between two CVRP solutions. As their code is not available, we opt to use their reported results directly.

### E.2 Generalization on real-world datasets

We further evaluate the generalization performance of our NeuOpt models, which were trained on TSP100 and CVRP100, by testing them on real-world TSPLIB and CVRPLIB instances containing up to 200 nodes (customers). These real-world instances differ significantly from our training ones

Table 10: Generalization results of our NeuOpt (10 runs) on TSPLIB real-world dataset.

| Instances | AM-mix | POMO-mix | AMDKD (POMO) | DACT (sol. = 10k) Avg. | Best | Ours (sol. = 10k) Avg. | Best | DACT (sol. = 40k) Avg. | Best | Ours (sol. = 25k) Avg. | Best |
|---|---|---|---|---|---|---|---|---|---|---|---|
| KroA100 | 4.02% | 0.02% | 0.02% | 1.13% | 0.41% | 0.00% | 0.00% | 0.21% | 0.00% | 0.00% | 0.00% |
| KroB100 | 4.73% | 0.25% | 0.41% | 0.67% | 0.26% | 0.05% | 0.00% | 0.38% | 0.26% | 0.00% | 0.00% |
| KroC100 | 7.60% | 0.01% | 0.02% | 2.56% | 0.39% | 0.00% | 0.00% | 0.77% | 0.00% | 0.00% | 0.00% |
| KroD100 | 8.45% | 0.27% | 0.09% | 2.23% | 0.45% | 0.05% | 0.00% | 0.64% | 0.45% | 0.02% | 0.00% |
| KroE100 | 3.61% | 0.50% | 0.53% | 1.14% | 0.46% | 0.02% | 0.00% | 0.60% | 0.28% | 0.02% | 0.00% |
| eil101 | 5.45% | 1.90% | 2.55% | 2.23% | 1.59% | 0.02% | 0.00% | 1.75% | 1.27% | 0.00% | 0.00% |
| lin105 | 17.29% | 0.36% | 0.36% | 6.37% | 2.69% | 0.00% | 0.00% | 0.89% | 0.00% | 0.00% | 0.00% |
| pr107 | 71.89% | 0.78% | 1.62% | 4.33% | 2.87% | 1.33% | 0.81% | 4.07% | 2.87% | 0.70% | 0.54% |
| pr124 | 5.16% | 0.00% | 0.43% | 0.75% | 0.65% | 0.12% | 0.00% | 0.30% | 0.00% | 0.05% | 0.00% |
| bier127 | 133.13% | 0.80% | 0.65% | 3.39% | 2.01% | 0.77% | 0.51% | 2.43% | 1.71% | 0.59% | 0.32% |
| ch130 | 1.98% | 0.60% | 0.69% | 2.72% | 0.41% | 0.40% | 0.29% | 1.09% | 0.41% | 0.35% | 0.29% |
| pr136 | 3.54% | 1.76% | 1.49% | 5.65% | 1.45% | 1.06% | 0.26% | 4.58% | 0.58% | 0.80% | 0.02% |
| pr144 | 13.82% | 0.85% | 0.72% | 2.63% | 2.07% | 0.62% | 0.17% | 1.16% | 0.44% | 0.19% | 0.03% |
| ch150 | 2.33% | 0.70% | 1.25% | 1.77% | 0.58% | 0.36% | 0.00% | 1.23% | 0.87% | 0.26% | 0.00% |
| KroA150 | 11.22% | 0.75% | 1.07% | 4.21% | 1.74% | 0.30% | 0.00% | 5.15% | 0.66% | 0.11% | 0.00% |
| KroB150 | 9.39% | 0.78% | 0.76% | 3.22% | 1.64% | 0.95% | 0.57% | 2.45% | 1.64% | 0.55% | 0.15% |
| pr152 | 16.31% | 1.35% | 2.16% | 3.10% | 1.57% | 0.79% | 0.34% | 2.23% | 0.77% | 0.26% | 0.00% |
| rat195 | 39.35% | 4.25% | 4.68% | 3.92% | 2.02% | 2.40% | 1.42% | 3.28% | 2.02% | 2.02% | 1.25% |
| KroA200 | 16.77% | 1.61% | 1.83% | 5.43% | 3.05% | 2.18% | 1.67% | 3.01% | 2.33% | 1.59% | 1.39% |
| KroB200 | 15.75% | 0.77% | 2.36% | 8.27% | 5.46% | 5.52% | 3.88% | 6.39% | 5.23% | 4.07% | 2.93% |
| Avg. Gap | 19.59% | 0.92% | 1.18% | 3.29% | 1.59% | **0.85%** | **0.50%** | 2.13% | 1.09% | **0.58%** | **0.35%** |

with much larger sizes and different node (customer, depot) distributions. We compare our NeuOpt with the state-of-the-art L2S solver, DACT [9], reporting both the best and average gaps across 10 independent runs. We also compare with the results from a recent work [11] that focused on enhancing the generalization capabilities of existing L2C solvers, where we include AM-mix, POMO-mix, and AMDKD (POMO) as baselines. According to Bi et al. [11], AM-mix and POMO-mix are enhanced AM and POMO models, respectively, which were trained on a combination of uniform, cluster, and a mix of uniform and cluster instances to bolster their generalization performance. The AMDKD approach leverages adaptive multi-distribution knowledge distillation to glean various forms of knowledge from multiple teachers trained on exemplar distributions, so as to yield a lightweight yet highly adaptable student POMO model. Since our primary focus is on the generalization capability of the deep model itself, we do not consider post-hoc per-instance processing boosters (e.g., EAS [5]). The results for TSPLIB and CVRPLIB are presented in Table 10 and Table 11, respectively.

**TSPLIB results.** As depicted in Table 10, our NeuOpt model consistently yields lower average gaps than the state-of-the-art DACT solver when both are allowed to explore 10k solutions. Meanwhile, our NeuOpt exploring 25k solutions could even surpass the DACT exploring 40k solutions. When compared to L2C solvers, NeuOpt yields lower gaps than AM-mix and POMO-mix, and even the AMDKD (POMO) model that is explicitly designed to enhance cross-distribution generalization performance. These results indicate the desired generalization capabilities of our NeuOpt model.

**CVRPLIB results.** As depicted in Table 11, our NeuOpt could still deliver the lowest average gaps on CVRPLIB instances compared to other baselines. Despite the promising results, we acknowledge that there is still room for improvement in enhancing the generalization of L2S solvers (including our NeuOpt), which is akin to the efforts made by AMDKD in advancing L2C solvers. Meanwhile, as discussed in Section 7, the integration of post-hoc per-instance processing boosters (e.g., EAS [5]) could potentially further improve both the in-distribution and cross-distribution generalization performance of existing L2S solvers. We consider the above ideas as important directions for future research.

### E.3 Results on TSP-200 and CVRP-200

We carry out experiments on larger scales, i.e., training our NeuOpt on TSP200 and CVRP200. For the size-specific hyper-parameters, we use $\xi = 0.125$ for both problems; and use the CVRP problem settings: capacity $\Delta = 70$ and 30 dummy depots. All other hyper-parameters remain unchanged. The performance of our NeuOpt models for TSP-200 and CVRP-200 are depicted in Table 12, where we compare our models with traditional heuristics (as per Table 1). The results indicate that our models consistently find close-to-optimal solutions, even outperforming the LKH3 solver on CVRP-200. We note that due to prohibited longer training times, existing L2S solvers (like DACT [9]) may not be efficient for training on size 200, which underscores the better scalability of our approach.

Table 11: Generalization results of our NeuOpt-GIRE (10 runs) on CVRPLIB real-world dataset.

| Instances | AM-mix | POMO-mix | AMDKD (POMO) | DACT (sol. = 10k) | | Ours (sol. = 10k) | | DACT (sol. = 60k) | | Ours (sol. = 60k) | |
|---|---|---|---|---|---|---|---|---|---|---|---|
| | | | | Avg. | Best | Avg. | best | Avg. | Best | Avg. | best |
| X-n101-k25 | 9.92% | 10.58% | 6.19% | 3.11% | 1.68% | 3.40% | 1.77% | 1.86% | 1.32% | 1.84% | 0.51% |
| X-n106-k14 | 6.06% | 2.71% | 1.84% | 2.77% | 2.28% | 1.66% | 1.01% | 2.09% | 1.72% | 1.41% | 1.01% |
| X-n110-k13 | 4.66% | 1.36% | 2.30% | 2.38% | 0.37% | 2.26% | 0.98% | 1.10% | 0.00% | 1.11% | 0.27% |
| X-n115-k10 | 14.83% | 6.76% | 5.27% | 1.43% | 0.08% | 1.18% | 0.03% | 0.68% | 0.02% | 0.61% | 0.00% |
| X-n120-k6 | 20.71% | 4.99% | 2.04% | 6.01% | 3.49% | 2.34% | 0.89% | 4.00% | 2.71% | 0.95% | 0.52% |
| X-n125-k30 | 24.00% | 12.32% | 5.46% | 6.01% | 4.76% | 4.71% | 3.15% | 5.51% | 4.76% | 3.55% | 2.58% |
| X-n129-k18 | 6.54% | 2.27% | 1.76% | 5.28% | 3.98% | 2.84% | 1.64% | 3.91% | 3.48% | 2.03% | 1.26% |
| X-n134-k13 | 16.43% | 3.74% | 3.79% | 6.40% | 3.77% | 3.48% | 2.42% | 3.28% | 2.32% | 2.50% | 1.73% |
| X-n139-k10 | 10.03% | 3.41% | 2.69% | 3.05% | 1.65% | 3.85% | 2.39% | 1.13% | 0.53% | 2.28% | 0.99% |
| X-n143-k7 | 16.84% | 5.01% | 4.12% | 5.15% | 3.92% | 2.05% | 1.29% | 3.37% | 0.89% | 1.27% | 0.78% |
| X-n148-k46 | 42.24% | 22.48% | 8.16% | 4.75% | 3.61% | 6.30% | 4.70% | 4.24% | 2.60% | 4.66% | 2.96% |
| X-n153-k22 | 16.50% | 11.35% | 11.17% | 7.62% | 4.63% | 9.16% | 6.89% | 4.26% | 3.22% | 6.71% | 6.00% |
| X-n157-k13 | 17.86% | 6.36% | 3.40% | 3.36% | 2.70% | 3.50% | 2.71% | 2.60% | 2.09% | 2.67% | 2.33% |
| X-n162-k11 | 4.41% | 5.75% | 5.41% | 2.68% | 1.82% | 3.02% | 1.92% | 1.57% | 1.10% | 2.43% | 1.82% |
| X-n167-k10 | 5.49% | 4.94% | 4.11% | 5.75% | 4.36% | 5.55% | 3.64% | 4.83% | 4.05% | 4.34% | 3.04% |
| X-n172-k51 | 36.86% | 9.29% | 9.06% | 5.42% | 3.66% | 8.85% | 5.92% | 3.78% | 3.37% | 6.48% | 5.43% |
| X-n176-k26 | 11.40% | 13.25% | 11.25% | 10.91% | 7.74% | 11.20% | 8.00% | 9.13% | 7.74% | 7.81% | 5.86% |
| X-n181-k23 | 6.83% | 13.47% | 4.64% | 3.65% | 3.05% | 2.82% | 2.19% | 2.51% | 1.89% | 2.33% | 2.07% |
| X-n186-k15 | 7.04% | 6.97% | 9.06% | 8.01% | 6.80% | 5.54% | 3.45% | 6.10% | 5.01% | 4.71% | 4.01% |
| X-n190-k8 | 23.79% | 11.38% | 3.50% | 7.86% | 6.36% | 8.06% | 6.18% | 6.42% | 5.81% | 6.65% | 5.76% |
| X-n195-k51 | 30.76% | 10.59% | 15.96% | 7.74% | 6.60% | 9.11% | 7.53% | 6.25% | 5.14% | 7.35% | 6.12% |
| Avg. Gap | 15.87% | 8.05% | 5.77% | 5.21% | 3.68% | **4.80%** | **3.27%** | 3.74% | 2.85% | **3.51%** | **2.62%** |

Table 12: Results of our NeuOpt approach on solving TSP-200 and CVRP-200 instances.

| Methods | Model Type | TSP-200 | | | CVRP-200 | | |
|---|---|---|---|---|---|---|---|
| | | Obj. | Gap | Time | Obj. | Gap | Time |
| Concorde [54] | Exact | 10.687 | - | 23m | | - | |
| HGS [21] | Heuristics | | - | | 21.756 | - | 19.8h |
| LKH [20, 51] | Heuristics | 10.687 | 0.00% | 2.3h | 22.010 | 1.17% | 21.6h |
| Ours (DA=5, T=1k) | L2S/RL | 10.732 | 0.42% | 28m | 22.214 | 2.11% | 58m |
| Ours (DA=5, T=5k) | L2S/RL | 10.696 | 0.09% | 2.4h | 21.960 | 0.94% | 4.8h |
| Ours (DA=5, T=10k) | L2S/RL | 10.692 | 0.04% | 4.7h | 21.904 | 0.68% | 9.6h |
| Ours (DA=5, T=20k) | L2S/RL | 10.689 | 0.02% | 9.4h | 21.861 | 0.48% | 19.2h |
| Ours (DA=5, T=30k) | L2S/RL | 10.688 | 0.01% | 14.1h | 21.842 | 0.39% | 1.2d |
| Ours (DA=5, T=40k) | L2S/RL | 10.688 | 0.01% | 18.8h | 21.830 | 0.34% | 1.6d |

## E.4 Inference with multiple GPUs

Our implementation allows for accelerated inference using multiple GPUs. Recall that in Table 1, the recorded time is for solving a total of 10,000 instances on a single GPU. For practical applications, users can achieve significantly shorter runtimes using multiple GPUs, as illustrated in Table 13.

Table 13: Inference time of our NeuOpt for solving 10k instances using different numbers of GPUs.

| Methods | TSP-100 | | | CVRP-100 | | |
|---|---|---|---|---|---|---|
| | 1GPU (11GB) | 2GPU (22GB) | 4GPU (44GB) | 1GPU (11GB) | 2GPU (22GB) | 4GPU (44GB) |
| NeuOpt (DA=1,T=1k) | 17m | 8m | 5m | 28m | 14m | 8m |
| NeuOpt (DA=1,T=5k) | 1.4h | 42m | 23m | 2.3h | 1.2h | 36m |
| NeuOpt (DA=1,T=10k) | 2.8h | 1.4h | 45m | 4.6h | 2.3h | 1.2h |

## E.5 Visualization of exploration behaviour

Figure 12 presents the objective values (search trajectories) of the visited feasible or infeasible solutions within the first 50 search steps, when using a pre-trained NeuOpt-GIRE model to solve five randomly generated CVRP-20 instances, respectively. The blue line depicts the objective values of each visited solution where we use green dots to mark feasible solutions and use red crosses to mark infeasible ones. The red and green dotted lines represent the best-so-far infeasible and feasible objective values, respectively. As observed from the figure, our learned model exhibits a propensity to explore infeasible regions at the beginning of the search, so as to rapidly decrease the objective values. It then learns to correct the constraint violations on the found best-so-far infeasible solutions and begins exploring the feasible regions, leading to new best-so-far feasible solutions more

efficiently. Moreover, there is a trend of alternating searches between feasible and infeasible regions. This behaviour aligns with our motivation in Section 5, suggesting that exploring infeasible regions could foster shortcut discovery, promising boundary searches, and identification of possibly isolated feasible regions. This showcases the significance and effectiveness of our GIRE method.

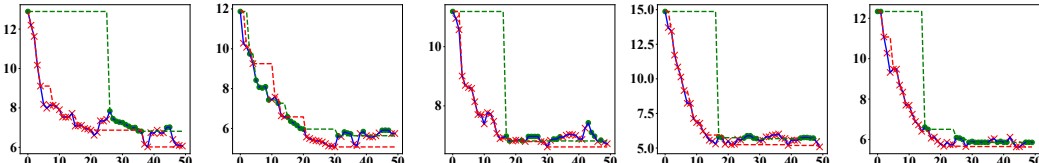

Figure 12: Search trajectories (50 steps) on 5 random CVRP-20 instances. The blue line shows the objective values of each visited solution (green dot - feasible one; red cross - infeasible one). The red and green dotted lines represent the best-so-far infeasible and feasible objective values, respectively.

In Figure 13, we further visualize the estimated distribution of the number of infeasible solutions within the last 5 visited solutions before finding a new best-so-far feasible solution. Results suggest that the probability of encountering at least one infeasible solution before finding a better best-so-far solution is around 80%, showcasing the importance of exploring both feasible and infeasible regions by GIRE (compared to purely visiting feasible solutions in the masking scheme).

| | P(#ifeasi. is 5/5) | P(#ifeasi. is 4/5) | P(#ifeasi. is 3/5) | P(#ifeasi. is 2/5) | P(#ifeasi. is 1/5) | P(#ifeasi. is 0/5) |
|---|---|---|---|---|---|---|
| T in [0.0k,0.2k) | 0.11 | 0.15 | 0.19 | 0.22 | 0.20 | 0.14 |
| T in [0.2k,0.4k) | 0.07 | 0.12 | 0.19 | 0.21 | 0.22 | 0.19 |
| T in [0.4k,0.6k) | 0.05 | 0.11 | 0.20 | 0.24 | 0.21 | 0.20 |
| T in [0.6k,0.8k) | 0.05 | 0.16 | 0.16 | 0.18 | 0.23 | 0.21 |
| T in [0.8k,1.0k) | 0.06 | 0.10 | 0.17 | 0.25 | 0.21 | 0.19 |

Figure 13: Estimated distribution of the number of infeasible solutions explored in 5 visited solutions before finding a new best-so-far feasible solution (as step size T increases).

## E.6 Used assets and licenses

We list the used assets in Table 14. All of them are open-source for academic research use. Our code and pre-trained models have been made available on GitHub (https://github.com/yining043/NeuOpt) using the MIT License.

Table 14: List of used assets (pre-trained models, codes, and datasets).

| Type | License | Asset |
|---|---|---|
| Code | MIT license | HGS [21], GCN+BS [14], DPDP [42], AM+LCP* [33], Pointerformer [32], Sym-NCO [13], POMO [4], POMO+EAS [5], POMO+EAS+SGBS [34], DACT [9], DIFUSCO [15] |
| | BSD-3-Clause license | Concorde [54] |
| | GPL-3.0 license | NLNS [8] |
| | Available for academic use | LKH-2 [51], LKH-3 [20] |
| | No license | Costa et al. [16], Wu et al. [39] |
| Dataset | Available for academic use | TSPLIB (http://comopt.ifi.uni-heidelberg.de/software/TSPLIB95/), CVRPLIB (http://vrp.galgos.inf.puc-rio.br/index.php/en/) |

