# OpenReview forum: "Learning to Search Feasible and Infeasible Regions of Routing Problems with Flexible Neural k-Opt"
_NeurIPS.cc/2023/Conference — NeurIPS 2023 poster_

### Official Review · Reviewer_2dzf · 2023-07-03

**Soundness:** 2 fair
**Presentation:** 3 good
**Contribution:** 2 fair
**Rating:** 5
**Confidence:** 4

**Summary:**

The paper introduces a learning-to-search (L2S) solver called Neural k-Opt (NeuOpt) for routing problems. NeuOpt learns to perform flexible k-opt exchanges using a tailored action factorization method and a customized recurrent dual-stream decoder. The paper also proposes the Guided Infeasible Region Exploration (GIRE) scheme, which allows the autonomous exploration of both feasible and infeasible regions. The experiments on TSP and CVRP with up to 100 nodes show that NeuOpt could outperform other learning-based methods slightly. However, the solving speed could be very slow. It also cannot outperform HGS and LKH3 from OR fields. Overall, I think the topic and proposed method are interesting, however, the results are not significant.

**Strengths:**

* The paper is mostly well-written, except that Figure 1 is too complex to help the reviewer to understand the basic ideas of the proposed method.
* The literature reviews are quite impressive, which include necessary classic and SoTA methods.
* The proposed NeuOpt and Guided infeasible region exploration seem reasonable.
* The proposed MDP for k-opt is interesting

**Weaknesses:**

* The experiments on TSP and CVRP with up to 100 nodes show that NeuOpt could outperform other learning-based methods slightly.  However, the solving speed could be very slow. It also cannot outperform HGS and LKH3 from OR fields. Overall, I think the results are not significant.
* Given the complexity of the method and the code is not provided, reproducibility could be a problem.

**Questions:**

The feasibility of solutions is not discussed in the experiments.  To what extent does the Guided Infeasible Region Exploration method impact the feasibility?  The reviewer encourages the authors to evaluate the method on VRPTW or TSPTW to make a more solid benchmark.

**Limitations:**

The limitations have been discussed in the main body. In Section 7,  the first point "it falls short against some L2P solvers (e.g., [5, 6]) for larger-scale TSPs" is a good discussion about the limitation. However, the second and third points are actually the advantages of the paper, rather than the limitations.

---

> ### Author Rebuttal · Authors · 2023-08-08
>
> We thank the reviewer for acknowledging that our approach is reasonable, the MDP is interesting, and the paper is mostly well-written. We understand that the main concerns are the significance of the performance and the code availability. We hope our response below will clarify any misunderstandings and concerns about our work.
> ***
> **[Significance of the Performance]** Regarding - *"NeuOpt could outperform other learning-based methods slightly. However, the solving speed could be very slow. It also cannot outperform HGS and LKH3 from OR fields"* - **these appear to be misunderstandings**, and we would like to clarify the following:
> * **Compared to learning-to-search (L2S) solvers: NeuOpt significantly surpasses them in both performance and speed** (NeuOpt also belongs to L2S solvers)
>     * On TSP100, our NeuOpt (0.33%, 17m) **halves the gaps** of Costa et al. (0.77%, 1.1h), Sui et al. (0.74%, 1.3h), and Wu et al. (1.54%, 2h) **with faster speed**. And NeuOpt (0.00%, 7h) **significantly surpasses** the SOTA solver DACT (0.10%, 13.5h).
>     * On CVRP100, our NeuOpt (0.85%, 2.3h) **reduces by an order of magnitude the gaps achieved by all L2S solvers with faster speed**, including NLNS (2.26%, 2.4h), NCE (1.59%, 10.4d), Wu et al. (3.87%, 5h), and DACT (1.11%, 1.7d).
> * **Compared to learning-to-construct (L2C) solvers: NeuOpt consistently achieves better/comparable performance with faster speed**
>     * On TSP100, our NeuOpt (0.02%, 2.8h) surpasses all L2C solvers **in both gap and speed**, including AM+LCP (0.60%, 10.9h), Pointerformer (0.11%, 5.6h), Sym-NCO (0.08%, 5.6h), POMO (0.07%, 5.6h), POMO+EAS (0.05%, 10.9h), and POMO+EAS+SGBS (0.03%, 1.1d).
>     * On CVRP100, NeuOpt (0.59%, 4.6h) surpasses L2C solvers like Sym-NCO (0.89%, 7.2h) and POMO (0.70%, 7.2h) **in both gap and speed**. Compared to POMO+EAS, to achieve gap 0.30%, NeuOpt is around **2 hours faster**. Compared to POMO+EAS+SGBS, to achieve gap 0.10%, NeuOpt is around **7 hours faster**.
>     * We note that POMO+EAS+SGBS has evolved through 4 stacked schemes, including model design by AM (ICLR’18), training algorithms by POMO (NeurIPS’20), active search by EAS (NeurIPS’21), and beam search by SGBS (ICLR’22). Thus, **it is already promising that our NeuOpt, utilizing unified model parameters for all test instances without per-instance active search (model parameter update) or beam search boosters, outperforms POMO+EAS+SGBS**. The superiority of our NeuOpt could be further enhanced with similar per-instance boosters in the future.
> * **Compared to learning-to-predict (L2P) solvers: NeuOpt shows greater adaptability to constrained VRPs and still exhibits better performance on the whole**
>     * On TSP100, our NeuOpt (0.33%, 17m) surpasses GCN+BS (1.35%, 46m), CVAE-Opt-DE (0.34%, 1.8d), and GNN+GLE (0.58%, 2.8h) in **both gap and speed**. Our NeuOpt (0.00%, 7h) surpasses the SOTA solver DIFUSCO (0.02%, 21.7h) with a **significantly faster speed**.
>     * While we acknowledge that NeuOpt does not outperform DPDP and Att-GCN+MCTS on TSP100, **their performance is limited to TSP only**. On CVRP100, our NeuOpt (0.30% gap, 13.8h) significantly surpasses DPDP (0.41% gap, 1.2d) and Att-GCN+MCTS even fails to solve CVRP.
> * **Compared to solvers from OR fields: our NeuOpt exhibits lower gaps than LKH3 with faster speed on both CVRP100 and CVRP200**
>     * On CVRP100, in Table 1 (main paper), **NeuOpt (0.30%, 13.8h) outperforms LKH3 (0.54%, 5.7day)**.
>     * On CVRP200, In Table 5 (appendix), **NeuOpt (0.68%, 9.6h) outperforms LKH3 (1.17%, 21.6h)**.
>     * We acknowledge that NeuOpt may not outperform the upgraded HGS solver recently released in 2022 (neither can all the other neural solvers given that neural solvers are still in early stages). Nevertheless, we narrow such gaps for this line of neural solvers. Moreover, by integrating per-instance search boosters like EAS, our neural solver has the potential to further amplify the performance.
>
> **Lastly, beyond mere performance competition, we present novel ideas and insights as recognized by other reviewers.** We believe that our contributions, including the k-opt factorization in MDP, the first L2S solver for flexible k-opt, the fresh constraint handling scheme GIRE, and the dynamic data augmentation are worth sharing with the learning-to-optimize community.
> ***
> **[Code availability]** As promised in lines 278-279, **we will make our code, pre-trained models, and the used data publicly available on GitHub. Following the rebuttal guidelines, we have forwarded our code to Area Chair**. Meanwhile, we note that our approach is not complex. For training on CVRP, it requires only 4 days, 5 days, and 8 days for sizes 20, 50, and 100, which are highly desirable as they are around half the time taken by POMO and DACT (their training time is around 2 weeks on CVRP100).
> ***
> **[Refine Figure 1]** Thanks for the valuable feedback. We will enhance the readability of Figure 1 and add explanations. Please refer to the global response for detailed clarification.
> ***
> **[Feasibility of the solution in the experiments]** We apologize for the confusion. The eventual solutions in all experiments are **always feasible** (because our approach only retrieves the best feasible solution visited during search as the final output), even though we allow temporal exploration of infeasible regions. Kindly refer to Appendix E.3 and Figure 9 for visualizations of the learned search trajectories (showing a trend of alternating searches between feasible and infeasible regions).
> ***
> **[Make benchmark more solid]** Thanks for the suggestion. We will study the extensions of NeuOpt-GIRE to more constraints in the future. However, we believe that the current benchmark is considered solid as recognized by other reviewers.
> ***
> **[Discuss limitations]** We apologize for the confusion. The last two points are future works. We will refine our paper to ensure all limitations and suggestions from reviewers are mentioned and addressed properly.

---

> > ### Comment · Area_Chair_92YU · 2023-08-18
> > **Thanks for authors' rebuttal!**
> >
> > Reviewer 2dzf, did the authors address your concerns about significance of the performance, code availability and other concerns? Thanks.

---

> > ### Comment · Reviewer_2dzf · 2023-08-19
> > **Thank you for the response**
> >
> > I truly appreciate the authors for taking the time to address my concerns.
> >
> > Upon thorough consideration of all responses including mine and others, I maintain my viewpoint that the results, in comparison to baseline LKH-3 and HGS, still lack significance. This is especially noteworthy given the relatively small scale of the routing problem and the time required for solving (spanning hours and days). While I acknowledge that the proposed method could potentially be integrated into approaches like TAM (Two-stage Dividing Method) [1] and L2D (Learn-to-Delegate) [2] to tackle larger-scale VRPs as sub-solvers, it's concerning that the execution time would considerably increase when employing the proposed method as opposed to utilizing LKH-3 or HGS (could obtain good results fast for small-scale VRPs).
> >
> > The discussion about the feasibility and extension to other constraints still has a lot of room to improve.
> >
> > An additional point of consideration is the method employed to measure the solving time of the proposed approach, as well as the corresponding benchmarks like LKH-3 and HGS. Clear elaboration on this matter is necessary.
> >
> > Despite the aforementioned lingering concerns, I do find the proposed ideas interesting, particularly the MDP formulation for k-opt. I am also anticipating the release of the code.
> >
> > On the whole, I am inclined to raise and finalize my evaluation score to 5, with the expectation that the outlined matters will be duly addressed and clarified.
> >
> > [1] Hou Q, Yang J, Su Y, et al. Generalize Learned Heuristics to Solve Large-scale Vehicle Routing Problems in Real-time[C]//The Eleventh International Conference on Learning Representations. 2023.
> >
> > [2] Li S, Yan Z, Wu C. Learning to delegate for large-scale vehicle routing[J]. Advances in Neural Information Processing Systems, 2021.

---

> > > ### Author Response · Authors · 2023-08-19
> > > **Thank you for the support and please see our responses to outlined matters (1/2)**
> > >
> > > We deeply appreciate the reviewer for considering raising the score. Thank you for acknowledging that our paper introduces interesting ideas, particularly the k-opt MDP formulation (besides presenting the first flexible k-opt solver, we also rethink the constraint handling by proposing GIRE and present the effective RDS decoder as well as the efficient dynamic data augmentation method). Below we further respond to your outlined matters.
> > >
> > > Regarding the solving time detailed in Table 1, we clarify that while our approach did take a long run time for certain cases, such long run time may be exclusive to the commonly used benchmarking setting, i.e., solving a total of 10,000 instances using one GPU only. Hence,  all the compared neural solvers share similar long run times (e.g., see the hours and days run times of SOTA baselines DACT and SGBS in our paper and their original papers). Given the limited memory of one GPU (e.g., 11GB for our 2080TI GPU), we need to split all 10,000 instances into smaller batches (e.g., 2000 instances) and run the batch inference in sequential (thus longer run time). **For practical use of our NeuOpt, if users have the flexibility to use multiple GPUs or more powerful GPUs (like the A100 with 80GB memory) or even TPUs, the runtime could be significantly reduced as shown in the added Table below.** Meanwhile, as mentioned by the Reviewer #vydT, the users can choose proper K, T and DA according to the available computation budgets. Lastly, we note that one of the motivations of learning to search (L2S) solvers is to close the optimality gaps as much as possible given enough run time, and our NeuOpt has significantly enhanced the efficiency of existing L2S solvers.
> > >
> > > --
> > >
> > > Time on TSP100|1GPU (11GB)|2GPU (22GB)|4GPU (44GB)
> > > :-:|:-:|:-:|:-:
> > > NeuOpt (DA=1,T=1k)|17m|8m|5m
> > > NeuOpt (DA=1,T=5k)|1.4h|42m|23m
> > > NeuOpt (DA=1,T=10k)|2.8h|1.4h|45m
> > > NeuOpt (DA=5,T=1k)|1.4h|43m|21m
> > > NeuOpt (DA=5,T=3k)|4.2h|2.2h|1h
> > > NeuOpt (DA=5,T=5k)|7h|3.6h|1.7h
> > >
> > > --
> > >
> > > Time on CVRP100|1GPU (11GB)|2GPU (22GB)|4GPU (44GB)
> > > :-:|:-:|:-:|:-:
> > > NeuOpt (DA=1,T=1k)|28m|14m|8m
> > > NeuOpt (DA=1,T=5k)|2.3h|1.2h|36m
> > > NeuOpt (DA=1,T=10k)|4.6h|2.3h|1.2h
> > > NeuOpt (DA=5,T=6k) |13.8h|7h|3.3h
> > > NeuOpt (DA=5,T=20k)|1.9d|22h|11h
> > > NeuOpt (DA=5,T=40k)|3.8d|1.8d|22h
> > >
> > > --
> > >
> > > We follow the recognized benchmark conventions (i.e., also used in the latest DACT, EAS, SBGS, and DIFUSCO papers) that the run time is recorded under the premise that one GPU is used for neural solvers and one CPU is used for traditional solvers. Nevertheless, we acknowledge the inherent challenges of time comparison between neural solvers and traditional solvers given the differences in infrastructure (CPU vs GPU) and programming languages (C vs Python), as mentioned in lines 291-297. We will make this more clear following the suggestions. Compared to LKH-3, our NeuOpt has shown promising performance by achieving lower optimality gaps with relatively shorter run times. And kindly note that our NeuOpt is the first L2S solver to achieve so. We acknowledge that our NeuOpt may not fully outstrip SOTA traditional solvers (i.e., LKH and HGS) that have been developed for more than decades. However,  this may also be the case for all existing neural solvers proposed in recent years. We note that our motivation is to unleash the potential of L2S solvers to further push the boundaries of neural solvers. Meanwhile, if we consider the idea of *“No Free Lunch”*, it is fair that no solver could be the best in all situations. We note that our approach does exhibit unique advantages compared to LKH-3 or HGS where our NeuOpt is able to learn and leverage deep patterns directly from data and rely less on domain knowledge regarding the human expertise of the target VRP, thus holding the potential to be swiftly adapted to automatically learn-to-solve more VRP variants (i.e., a generic tool to learn data-driven VRP solvers).

---

> > > > ### Author Response · Authors · 2023-08-19
> > > > **Thank you for the support and please see our responses to outlined matters (2/2)**
> > > >
> > > > We thank the reviewer for acknowledging that our NeuOpt could potentially be integrated into divide-and-conquer frameworks for large-scale VRPs. We will elaborate further on this and highlight the TAM and L2D frameworks. To clarify, we believe that using NeuOpt as sub-problem solver may not induce extensive run time, given that the sub-problem can be solved in parallel and more GPUs can be used. Meanwhile, we can adopt reduced inference steps (T) while expanding the number of data augmentations (DA), with all augmented instances being solved in parallel. We also note that our training time is much less than POMO and DACT (our 8 days compared to their 2 weeks). **Moreover, compared to LKH-3 or HGS, leveraging NeuOpt as conquer (sub)-solvers brings several unique benefits:**
> > > > * **End-to-End Training:** Integrating trainable divider like TAM with trainable conquer solvers like our NeuOpt enables end-to-end training of the bi-level framework. Following this advantage, future works may promote the exchange of contextual information between the neural divider and the neural conquer solver for better performance.
> > > > * **Batch Training/Inference:** Unlike traditional CPU-based solvers which often necessitate sequential solving of a batch of instances, our GPU-based NeuOpt allows for better parallelization and promotes larger batch size for training/inference.
> > > > * **Do Not Require Pre-Existing Solvers:**  Our NeuOpt diminishes dependency on pre-existing solvers. Users with no domain knowledge can first train a NeuOpt solver for the new VRP variant (especially when no existing solver exists), and then refine the trained NeuOpt model within a divide-and-conquer framework for solving larger-scale VRPs using DRL.
> > > >
> > > > **Our GIRE can be viewed as an important early attempt that moves beyond the pure feasibility masking to autonomously explore both feasible and infeasible regions during search.** In section 5, we discussed the motivations of GIRE at the beginning and provided the rationales of GIRE in *Feature Supplement* and *Reward Shaping* paragraphs. To adapt GIRE for a particular constraint, we suggest the following:
> > > > * For *Feature Supplement*, simply binary indicator functions can be employed to discern specific constraint violations of each node in the solution, thereby forming the Violation Indicator (VI) features. For example, VI can indicate nodes (customers) in the solution that breach their time window or the pickup/delivery precedence constraints. For the Exploration Statistics (ES) features, they can be retained as originally conceptualized since they are based on historical exploration records, rather than specific constraint.
> > > > * For *Reward Shaping*, it can be retained as originally conceptualized as well. Note that as discussed in Appendix D, lines 579 to 585, we recommend to characterize the $\epsilon$-feasible regions by a fraction of nodes (i.e., 10%) that do not adhere to the constraints.
> > > >
> > > > Regarding the application of GIRE to other constraints, we will try to implement and include them. Nevertheless, we believe that current evaluations about GIRE feasibility are already solid. In Section 6, we showed the application of GIRE to NeuOpt and to DACT in Table 2, and conducted GIRE ablation studies in Figure 4. In Appendix, we depicted discussions about GIRE including the choice of $\epsilon$ in Figure 7, the impact of reward shaping on search behavior in Figure 8, and the visualizations of the learned search trajectories in Figure 9. We will also add the discussion on the impact of GIRE entropy measures (see Figure A1 and A2 in the new PDF). Following the suggestions, we will further enrich the discussions. **Here, we provide an additional analysis on the feasibility of the visited solutions by GIRE on CVRP.** The table below offers a breakdown of the estimated probabilities associated with encountering a total of 0, 1, 2, 3, 4, or 5 infeasible solutions within the most recent 5 visited solutions prior to finding a better best-so-far solution. Results suggest that the probability of encountering at least one infeasible solution before finding a better best-so-far solution is around 80%, showcasing the importance of exploring both feasible and infeasible regions by GIRE (compared to purely visiting feasible solutions in the masking scheme).
> > > >
> > > > Search stages|P(# ifeas. 5/5)|P(# ifeas. 4/5)|P(# ifeas. 3/5)|P(# ifeas. 2/5)|P(# ifeas. 1/5)|P(# ifeas. 0/5)
> > > > :-:|:-:|:-:|:-:|:-:|:-:|:-:
> > > > T in [0k,0.2k)|0.11|0.15|0.19 |0.22|0.20 |0.14
> > > > T in [0.2k,0.4k)|0.07|0.12|0.19|0.21|0.22|0.19
> > > > T in [0.4k,0.6k)|0.05|0.11|0.20|0.24|0.21|0.20
> > > > T in [0.6k,0.8k)|0.05|0.16|0.16|0.18|0.23|0.21
> > > > T in [0.8k,1.0k)|0.06|0.10|0.17|0.25|0.21|0.19
> > > >
> > > > **Lastly, we are committed to releasing code on Github after the review process. We note that the Area Chair has kindly verified the receipt of our code repo link in a separate comment (might not be visible to reviewers).**
> > > >
> > > > We hope the above response clears your concerns.

---

### Official Review · Reviewer_vydT · 2023-07-06

**Soundness:** 3 good
**Presentation:** 3 good
**Contribution:** 3 good
**Rating:** 6
**Confidence:** 2

**Summary:**

The paper aims to learn the k-opt operation, one of the famous local search methods, via neural networks. In particular, the authors model the k-opt operation as a sequential node selection process and use a recurrent dual-stream (RDS) decoder. Furthermore, a guided infeasible region exploration (GIRE) is suggested to encourage the policy to escape local optima.
The paper models the k-opt as a sequence of basic moves on open Hamiltonian paths (not cycles), which allows the neural network model easily learns the k-opt operation. Since the end of the sequence (i.e., E-move) is a kind of I-moves, k can be flexibly adjusted according to states. GIRE allows the model to explore infeasible regions, not only feasible regions; it encourages the model to escape local optima. In addition, the portion of infeasible solutions is dynamically adjusted.

**Strengths:**

The paper is well-written in general, and the key ideas are clearly delivered. The extensive experiments with various baselines suggest that the proposed method can mitigate the inefficiency of L2S solvers.

One of the main benefits is that we can choose the proper K, the number of DA, and T according to the available computation budget. Also, the results show that NeuOpt gives promising performances with similar time with L2C algorithms.

**Weaknesses:**

There is no analysis of where the performances come from. the k-opt operations are well-working with fixed k. Comparisons between the original k-opt and NeuOpt are required.

Some detailed but important information is missing or hard to be found (e.g., encoding scheme, dynamic augmentation, initial tour)

**Questions:**

I have some questions about the details.

1. Effectiveness of learning the S-move. I think that choosing the S-move also gives good performances.
2. Why is LKH3 (known as better than LKH2) not employed as a baseline?
3. Whether the encoding is necessary for every t?
4. How is the problem augmented in detail? I’ve read through the appendix but couldn’t find it.

Minor Comments:
1. I think further explanation for ES features is required (maybe in the appendix)
2. For better readability, additional cross-references for contents in the appendix are recommended.
3. There is no explanation when the other papers are referenced (e.g., line 192, 257). I recommend adding further descriptions.

**Limitations:**

I think the dual-stream structure is restricted to consider additional context information.

---

> ### Author Rebuttal · Authors · 2023-08-08
>
> We appreciate the reviewer for the positive and valuable comments. Thank you for acknowledging that NeuOpt improves the efficiency of L2S solvers, exhibits unique benefits for practical use, and achieves promising performance. We hope that the following response, along with additional experimental results, will address the remaining concerns.
> ***
> **[Similar time with L2C solvers]** Thanks for the comment. However, we would like to clarify that **our NeuOpt is consistently faster than all compared L2C solvers** while achieving better or similar performance. Specifically,
> * On TSP100, our NeuOpt (0.02%, 2.8h) surpasses all L2C solvers **in both gap and speed**, including AM+LCP (0.60%, 10.9h), Pointerformer (0.11%, 5.6h), Sym-NCO (0.08%, 5.6h), POMO (0.07%, 5.6h), POMO+EAS (0.05%, 10.9h), and POMO+EAS+SGBS (0.03%, 1.1d).
> * On CVRP100, NeuOpt (0.59%, 4.6h) outperforms L2C solvers like Sym-NCO (0.89%, 7.2h) and POMO (0.70%, 7.2h) **in both gap and speed**. Compared to POMO+EAS, to achieve a gap of 0.30%, NeuOpt is around **2 hours faster**. Compared to POMO+EAS+SGBS, to achieve a gap of 0.10%, NeuOpt is around **7 hours faster**.
> ***
> **[Where does the performance come from?]** Thanks for the question. Following your suggestion, we will make the analysis more clear. We attribute the performance of our approach to our 4 contributions, thoroughly verified through extensive experiments and ablation studies in the original paper:
> * **New action factorization:** It enables autonomous scheduling of dynamic $k$ during search (see lines 40-42), delivering advantages over fixed $k$ (see Figure 5a, lines 349-356).
> * **The RDS decoder:** It is flexible to control k-opt with any $k$ and more effectively captures the strong correlations between the removed and added edges (see line 45). Compared to the DACT decoder, our RDS decoder achieves much better performance with reduced run time (see Table 2 and lines 329-331). Further, designs within the RDS decoder, including the GRUs and dual streams, are all essential to performance (see Table 3 and lines 333-336).
> * **The GIRE scheme:** It is the first constraint handling scheme that promotes the exploration of both feasible and infeasible regions, bringing multiple benefits (see lines 52-57). It has been verified to be generic to boost both DACT and our NeuOpt for better constraint handling (see Table 2, lines 326-328). Besides, designs within GIRE, including the reward shaping and feature supplement, are both essential to performance (see Figure 4 and lines 337-343). We also provide visualizations and discussions about GIRE (see Appendix D and Figure 8).
> * **Dynamic data augmentation:** It enables NeuOpt to explicitly escape from the local optima (see Table 4 and lines 344-348). Table 1 also shows the effects of different augmentation settings on performance.
> ***
> **[Comparison with other k-opt methods]** Following the suggestion, we have added another baseline called OriginOpt and gathered the results together with those reported in the original paper in the tables below. Note that various traditional and learning-based k-opt baselines are now comprehensively included, as detailed below:
> * **Traditional k-opt baselines:**
>     * **OriginOpt (static)** and **OriginOpt (dynamic)**, which randomly perform the k-opt (rather than using our NeuOpt) in a static and dynamic manner, respectively.
>     * **LKH**, which not only performs dynamic k-opt, but also integrates other complex heuristic designs for better performance (e.g., an $\alpha$-measure-based edge candidate set, partitioning rules, tour merging strategies, the iterative partial transcription technique, the backbone-guided search, etc).
> * **Learning-based k-opt baselines:**
>     * Neural 2-opt: **DACT**, **Wu et al.**, and **Costa et al.**
>     * Neural 3-opt: **Sui et al.**
>
> The results demonstrate that our NeuOpt significantly outperforms all learning-based k-opt baselines, as well as the original k-opt OriginOpt baseline. Notably, our NeuOpt is able to find lower gaps at a faster speed than the strong LKH3 solver (which employs complex heuristics beyond k-opt) on CVRP100.
>
> --
> TSP100|Gap|Time
> -|-|-
> LKH2|0.00%|5.7h
> OriginOpt (static)|210.00%|17m
> OriginOpt (dynamic)|202.40%|17m
> Costa et al.| 0.77%| 1.1h
> Sui et al.| 0.74%| 1.3h
> Wu et al.| 1.54%| 2h
> DACT| 0.10%| 13.5h
> **NeuOpt (DA=1,T=1k)**|0.33%|17m
> **NeuOpt (DA=5,T=3k)**|0.01%|4.2h
>
> --
>
> CVRP100|Gap|Time
> -|-|-
> LKH3 |0.54%|5.7d
> OriginOpt (static)|93.23%|2.3h
> OriginOpt (dynamic)|103.84%|2.3h
> Wu et al.|3.87%|5h
> DACT|1.11%|1.7d
> **NeuOpt (DA=1,T=5k)**|0.85%|2.3h
> **NeuOpt (DA=5,T=6k)**|0.30%|13.8h
> ***
> **[Effects of S-move]** Thanks for the insightful comment. The reviewer is correct that learning the S-move is crucial for good performance. Kindly refer to Figure A4 in the attached PDF under the global response for new results.
> ***
> **[LKH2 or LKH3]** Kindly note that **we employed LKH3 for CVRP, while LKH2 was used for TSP**. This distinction was made because LKH2 is the latest version for TSP, whereas the updates in LKH3 are only made for constrained TSP (such as CVRP).
> ***
> **[Encoding for every t?]** Yes, because the current solution and features change with each $t$. This is common across all existing L2S and L2C solvers.
> ***
> **[Is dual-stream structure restricted?]** No. Let $s_1, s_2$ be the logits of the current dual streams, and $s_3, s_4, …$ represent additional ones. While our paper utilizes $s_1 + s_2$, more streams can be easily accounted for by MLP($s_1, s_2, s_3, s_4, …$) if needed.
> ***
> **[Explain details & other minor comments]** Thanks for the suggestions. We will refine our paper accordingly. Specifically, encoding and augmentation details follow the exact methods used in [1]. We will include algorithm pseudocode for dynamic data augmentation, and we will clarify that the initial tours are randomly generated (same with all existing L2S solvers).
> ```
> References:
> [1] Efficient Neural Neighborhood Search for Pickup and Delivery Problems (IJCAI’22)
> ```

---

> > ### Comment · Reviewer_vydT · 2023-08-18
> > **Thank you for the responses.**
> >
> > Thank you for faithfully answering my responses. I will maintain my score.

---

> > > ### Author Response · Authors · 2023-08-18
> > > **Thank you for the support**
> > >
> > > We deeply appreciate the reviewer for acknowledging our response and keeping to support our paper. We will incorporate your valuable suggestions and our discussions in the revised paper.

---

### Official Review · Reviewer_8kc7 · 2023-07-12

**Soundness:** 3 good
**Presentation:** 3 good
**Contribution:** 3 good
**Rating:** 7
**Confidence:** 4

**Summary:**

In this paper, the authors propose Neural k-Opt (NeuOpt) that factorizes a generic k-opt exchange operation as a series of base operations. They also introduce Guided Infeasible Region Exploration (GIRE) scheme for Capacitated Vehicle Routing Problem (CVRP) where one augments a reward function for RL with signals about exploring infeasible solution spaces. On the model architecture side, the authors propose a recurrent dual-stream decoder that consumes both moves and edges. The authors provided a comprehensive comparison with baseline methods for problems of sizes up to 100 cities.


**Strengths:**

1. I found the action factorization novel and general. A limitation of past k-Opt based learning-to-search methods is that they require a pre-determined value of k. The proposed factorization method can accommodate arbitrary k given an agent enough number of steps.

2. GIRE seems a promising approach to incorporating information about near-feasible regions into explorations. As far as I know, this contribution is novel.

3. The empirical comparisons with baseline methods are comprehensive. I appreciate the authors including an up-to-date list of baseline methods. There are some expected issues with reproductions but I definitely applaud the authors’ efforts here.

4. The paper is relatively well-written. The authors did a good job of packing a dense set of information within the page limit.

**Weaknesses:**

1. The problem sizes considered are relatively small. Methods like DIFUSCO experimented with sizes up to 10000 cities. It is unclear how practically useful this method is with small problem sizes.

2. Another common evaluation mode for these classes of problems is for the generalization ability of a trained model. That is, extrapolate the model performance on problems of larger sizes than those in the training set. This is not considered in this paper.

3. In general, the empirical results are mixed. On TSP, NeuOpt is not better than DPDP and Att-GCN+MCTS. The comparison with DIFUSCO is obfuscated by a large time discrepancy from the original paper, possibly due to difficulty in reproducing their results. On CVRP, POMO+EAS+SGBS methods are comparable with NeuOpt.

4. The main advantage of the factorized action representation is that one could potentially handle arbitrary k-Opt in a general way. However, in the experiments, the authors used a value of 4 for k which seemed limiting. In addition, in the ablation on values of k, larger k values worked better, so why did the authors not choose a larger value of k for the main evaluations?

5. There are some places where the exposition could be improved. Please see my questions below.


**Questions:**

1. In section 5 about the reward shaping (near line 266), how to compute $P_t(\mathcal{U}|\mathcal{U})$ exactly?

2. In the entropy computation, why $P_t(\mathcal{U}|\mathcal{F})$ and $P_t(\mathcal{F}|\mathcal{U})$ are not included? It is possible to transition between feasible and infeasible solution spaces.

3. The actual entropy calculation involves hyperparameters $c_1$ and $c_2$. Can the authors provide more context on how they were chosen and their impact on the stability of training?

4. In Table 1, for different variants of NeuOpt, what does the DA number refer to? Is that the $T_{DA}$ mentioned in section 4.3?

5. In Figure 5(a), the results for K = 5 and 6 w/o E-move are missing.


**Limitations:**

The authors could add discussions about the generalization ability of trained models as well as scaling to larger problem sizes.

---

> ### Author Rebuttal · Authors · 2023-08-08
>
> We appreciate the reviewer for the positive and valuable comments. We are delighted that the reviewer found our approach novel, general, and promising. Thank you for recognizing our efforts to include extensive baselines. We hope that the following response, along with additional experimental results, will address the remaining concerns.
> ***
> **[How practically useful our NeuOpt is?]** Thanks for the comment.
> * Regarding - *"sizes considered are relatively small"* - we would like to clarify that we did improve the scalability of the learning-to-search (L2S) solvers by direct training on size 200 (see global response). For even larger instances, we recommend integrating our NeuOpt with the SOTA **divide-and-conquer** frameworks. In such frameworks (e.g., L2D [2] and RBG [3], references in global response), a divider is usually learned to segment large-scale instances (e.g., TSP with 10,000 nodes) into sub-problems (e.g., fifty TSP-200 instances), where the sub-problems are solved by LKH3 or POMO (in parallel). **Given the new SOTA performance on sizes 100 and 200, our NeuOpt could serve as a more desirable conquer solver in these frameworks**.
> * Regarding - *"methods like DIFUSCO experimented with sizes up to 10000 cities"* - we acknowledge that L2P solvers excel at scalability, however, **methods like DIFUSCO are limited to supervised learning and TSP only** since their predicted heatmaps may not handle other constraints. Conversely, our NeuOpt and GIRE could be more adaptable for constrained VRPs via DRL. Moreover, we have shown in Table 1 that our NeuOpt (0.00%, 7h) exhibited much better performance than DIFUSCO (0.02%, 21.7h) on TSP100.
> * Regarding - *"how practically useful this method is?"* - the unique **research and practical values** of our approach are listed as follows:
>     * SOTA performance compared to up-to-date baselines
>     * the first L2S solver for flexible k-opt search
>     * the first constraint-handling scheme beyond masking
>     * better conquer solver for divide-and-conquer frameworks
>     * more generic and adaptable solvers (for constrained VRPs)
> ***
> **[Generalization evaluation]** Yes, we have included such evaluations in the original Appendix. Kindly refer to the global response for more details.
> ***
> **[Time discrepancy with DIFUSCO]** Thanks for the question. Their reported time is for 128 instances while ours is for 10,000 instances. Meanwhile, their hardware (V100 GPU, CPU 2.50GHz) is different from ours (2080TI GPU, CPU 2.40GHz).
> ***
> **[Results are mixed]** We agree with the reviewer that different solvers have unique pros and cons, and we note that our NeuOpt does achieve SOTA performance on the whole. Specifically,
> * The good performance of **DPDP and Att-GCN+MCTS is limited to TSP, and our NeuOpt are more adaptable to constraints**. Our NeuOpt (0.30%, 13.8h) significantly outperforms DPDP (0.41%, 1.2d) on CVRP100 and Att-GCN+MCTS even fails to solve CVRP.
> * While achieving comparable gaps, **our NeuOpt-GIRE is around 7h faster than POMO+EAS+SGBS on CVRP100 and also significantly outperforms it on TSP100**. We note that POMO+EAS+SGBS has evolved through 4 stacked schemes, including model design by AM (ICLR’18), training algorithms by POMO (NeurIPS’20), active search by EAS (NeurIPS’21), and beam search by SGBS (ICLR’22). Thus, it is already promising that our NeuOpt, utilizing unified model parameters for all test instances without per-instance active search (model parameter update) or beam search boosters, outperforms POMO+EAS+SGBS. We believe the superiority of our NeuOpt could be further enhanced with similar per-instance boosters in the future.
> * Lastly, as one of the L2S solvers, **our NeuOpt is able to halve (on TSP) or even reduce by an order of magnitude (on CVRP) the gaps achieved by other L2S solvers with a much shorter run time**.
> ***
> **[Why use K=4?]** Thanks for the question. We use K=4 to balance computational costs and better performance, aligning with traditional solvers like LKH that avoid $k\geq5$ due to unbounded exploration. Yes, our factorization benefits from handling k-opt in a general way. Besides, it lets the model autonomously schedule dynamic k during the search. Kindly refer to our response **[Discuss trade-offs and computational costs]** and **[Why dynamic $k$ help escape local minima?]** to reviewer #NeSp for more discussions.
> ***
> **[Compute $P(\mathcal{U}|\mathcal{U})$]** We adopt $P(\tau'\in\mathcal{U}|\tau\in\mathcal{U})=\frac{P(\tau'\in\mathcal{U},\tau\in\mathcal{U})}{P(\tau\in\mathcal{U})}$, where $P(\tau'\in\mathcal{U},\tau\in\mathcal{U})$ and $P(\tau\in\mathcal{U})$ are estimated based on the historical solution records of the past $T_{EI}$ steps.
> ***
> **[Why not include $P(\mathcal{U}|\mathcal{F})$ and $P(\mathcal{F}|\mathcal{U})$?]** We opt not to explicitly include them since the MLP is able to derive $P(\mathcal{U}|\mathcal{F}) = 1 - P(\mathcal{F}|\mathcal{F})$ and $P(\mathcal{F}|\mathcal{U}) = 1 - P(\mathcal{U}|\mathcal{U})$.
> ***
> **[Effects of c1 & c2]** Thanks for the suggestion. They control the entropy measure patterns (shown in Figure 3). We set the values to only penalize extreme search behavior if the feasibility transition probability is outside [0.25, 0.75]. We follow the suggestion and add Figure A1 and Figure A2 in the attached PDF under global response. Results show that they may not affect training stability (thus no need for tuning in practical use).
> ***
> **[DA and T_DA]** We apologize for the confusion. DA refers to the number of augmentations for an instance, and T_DA is the maximum number of steps allowed before considering the search trapped in a local optimum. If T_DA is reached, the augmentation is changed to a new one. We will include algorithm pseudocode for dynamic data augmentation in the revised Appendix.
> ***
> **[Full results of w/o E-move]** Thanks for the comment. We have added the requested results to Figure A3 in the attached PDF under global response. The conclusion remains unchanged.

---

> > ### Comment · Reviewer_8kc7 · 2023-08-10
> > **Thank you for the response**
> >
> > I want to thank the authors for their detailed responses to my questions. I would suggest including the generalization results in the main paper in a revision since that is an important set of experiments showcasing the effectiveness of the proposed method. As my questions are well addressed, I will raise my score to 7.

---

> > > ### Author Response · Authors · 2023-08-11
> > > **Thank you for the support and suggestion**
> > >
> > > We deeply appreciate the reviewer for acknowledging our response. We will follow the suggestion to include the summarized Table A1 and Table A2 as well as corresponding discussions in the revised main paper.

---

### Official Review · Reviewer_NeSp · 2023-07-23

**Soundness:** 4 excellent
**Presentation:** 2 fair
**Contribution:** 3 good
**Rating:** 6
**Confidence:** 3

**Summary:**

The paper introduces Neural k-Opt (NeuOpt), a deep learning-based vehicle routing solver, designed to handle k-opt exchanges for any k≥2. Unlike existing Learning-to-Search (L2S) solvers, NeuOpt employs a tailored action factorization method, which allows complex k-opt exchanges to be broken down into simpler basis moves. This approach grants the model the flexibility to determine an appropriate k for each search step, enabling a balance between coarse-grained (larger k) and fine-grained (smaller k) searches.

Moreover, the paper introduces Guided Infeasible Region Exploration (GIRE) scheme that promotes exploration of both feasible and infeasible regions in the search space. GIRE enriches the policy network with additional features to indicate constraint violations and exploration behavior statistics, aiding in escaping local optima, discovering shortcuts to better solutions, and enhancing the model's understanding of the problem landscapes.

NeuOpt is trained using reinforcement learning (RL) and incorporates a dynamic data augmentation method during inference for improved efficiency. Extensive experiments on classic vehicle routing problem variants (TSP and CVRP) demonstrate the superiority of NeuOpt and GIRE over some existing approaches, including traditional hand-crafted solvers.

**Strengths:**

- Novel Approach: The paper introduces a novel approach called Neural k-Opt (NeuOpt) to handle k-opt exchanges for vehicle routing problems. This approach offers flexibility by allowing k to be any value ≥2, which can potentially lead to better solutions and more efficient search processes.

- Guided Infeasible Region Exploration (GIRE): The introduction of GIRE is a significant contribution that promotes exploration of both feasible and infeasible regions in the search space. This unique scheme bridges feasible regions, helps escape local optima, and forces explicit awareness of the VRP constraints, potentially leading to improved performance in constrained VRPs.

- Comprehensive Evaluation: The paper claims to have extensive experiments on classic VRP variants (TSP and CVRP) to validate the proposed NeuOpt and GIRE. A thorough evaluation of the model's performance on various VRP instances can demonstrate its effectiveness and potential advantages over existing methods.

**Weaknesses:**

- Scalability: While the paper claims that NeuOpt outperforms other methods, including hand-crafted solvers, there is no mention of its scalability to larger and more complex instances of vehicle routing problems. The effectiveness of NeuOpt on larger and real-world datasets should be explored.

- Complexity of Action Factorization: Although the tailored action factorization method is introduced to handle k-opt exchanges flexibly, it may introduce increased complexity in the model's architecture and training process. The paper should discuss potential trade-offs and computational costs associated with the proposed factorization.

**Questions:**

- Figure 1 is very hard to process...

- lines 122-123: I believe the definition of TSP is slightly inaccurate. The main objective of the TSP is to find the optimal Hamiltonian cycle that **minimizes** the total distance or cost required to visit all the nodes exactly once and return to the starting node.

- Can authors provide more discussions on why choosing k dynamically should escape local minimas?



**Limitations:**

see Weaknesses

---

> ### Author Rebuttal · Authors · 2023-08-08
>
> We appreciate the reviewer for the positive and valuable comments. We are delighted that the reviewer acknowledged that our contributions (NeuOpt and GIRE) are novel and significant, and our experiments are comprehensive and extensive. We hope that the following response will clear the remaining concerns.
> ***
> **[Effectiveness of NeuOpt on larger and real-world instances]** Thanks for raising the concern. We would like to clarify that our NeuOpt (for TSP) and NeuOpt-GIRE (for CVRP) were evaluated on these more complex instances as detailed in the original Appendix. Please refer to the above global response for a summary of these results.
> ***
> **[Are our action factorization complex?]** We understand the concern about complexity. As pointed out by the reviewer, our Action Factorization is designed to allow the deep model (our NeuOpt) to control k-opt exchanges flexibly (with any $k \ge 2$). We would like to clarify that our Action Factorization does not introduce much complexity into the model architecture or the training process. In fact, the designs of our NeuOpt for achieving flexible k-opt are both lightweight and desirable, for the following three reasons:
> * **Regarding the computational complexity:** Firstly, our factorization formulation is already simpler (while ensuring flexibility) compared to the original k-opt definition. Moreover, in the proposed RDS decoder, we **have further simplified** the parametrization of the above action factorization to a *node selection process* (with the type of basis moves in the action factorization being automatically inferred, see lines 198-201). This means that the decoder **only needs to specify one node in each decoding step**, resulting in a computational complexity (time) that grows approximately linearly with the increase of K. We believe that such complexity is highly desirable for decoding k-opt decisions.
> * **Regarding the model architecture:** We note that the model size (number of parameters) **remains the same for varying K** since our RDS decoder is flexible to control any k-opt with a unified model architecture. In Table 2 and our analysis in lines 329-331, we compared our NeuOpt (K=2) with DACT (2-opt) where both of them are learning to control 2-opt but with different decoders. Compared to the existing DACT decoder, our RDS decoder achieved better performance (our gap 0.24% vs its gap 0.32%) with **reduced** time (our 119s vs its 171s) but a slightly increased model size (our 0.683m vs its 0.633m).
> * **Regarding the training complexity:** We reported the training time in lines 615-617 in the original Appendix. For training on CVRP, it requires only 4 days, 5 days, and 8 days for sizes 20, 50, and 100, which are highly desirable as they are about **half the time taken by POMO and DACT** (their training time is around 2 weeks on CVRP100).
> ***
> **[Discuss trade-offs and computational costs]** We agree with the reviewer that there are trade-offs between better performance and increased computational costs. We would like to clarify that we did mention such trade-offs in the original submission:
> * In Table 2 and lines 329-331, we compared our NeuOpt (K=2) with NeuOpt (K=4), and mentioned that the performance is *"further amplified by increasing K to 4 for NeuOpt at the cost of slightly increased run time"*.
> * In Figure 5(a) and lines 354-357, we compared our NeuOpt with K=2, K=3, K=4, K=5, K=6, and mentioned that *"however, there is a trade-off between larger K and longer decoding time."*
>
> Following the suggestion of the reviewer, we have further **supplemented this by gathering more data to illustrate the trade-offs more clearly**. In the table below, we exhibit the performance and the inference/training time for different K. The results did suggest the aforementioned trade-offs, and we used K=4 to balance the trade-offs in this paper. We will further refine our paper to make the discussion more clear.
> | | Inference Time (T=1k) | Training Time (per epoch) | Objective Values | Gaps to the Best |
> |:-:|:-:|:-:|:-:|:-:|
> | NeuOpt (K=2) | 2m02s| 20m| 7.817| 0.30%|
> | NeuOpt (K=3) | 2m07s| 21m| 7.806| 0.15%|
> | NeuOpt (K=4) | 2m13s| 22m| 7.798| 0.05%|
> | NeuOpt (K=5) | 2m19s| 24m| 7.795| 0.01%|
> | NeuOpt (K=6) | 2m26s| 25m| 7.794| 0.00%|
> ***
> **[Refine Figure 1]** Thanks for the valuable feedback. We will enhance the readability of Figure 1 and add explanations. Please refer to the global response for detailed clarification.
> ***
> **[Refine TSP definition]** Thanks for the valuable feedback. We will refine it according to your suggestion and thoroughly check all the other definitions.
> ***
> **[Why dynamic $k$ help escape local minima?]** Thanks for the valuable question. We will refine our paper based on the discussions from two aspects as follows:
> * **Exploration of different neighborhoods:** The value of $k$ defines the neighbourhood of the current solution. For a given solution, different neighborhoods may have different local minima. For example, from the current solution $\tau_0$, we may obtain a local minima solution $\tau_1$ via a 2-opt move (only changing 2 edges of $\tau_0$), while a 5-opt move on $\tau_0$ might lead to another local minima solution $\tau_2$. Hence, the capability to vary $k$ dynamically can equip the solver with flexibility and reduce the likelihood of confinement to particular local minima.
> * **Adaptive search strategies:** Different search stages may require different $k$ to foster an efficient search. However, a statically defined $k$ might either be too restrictive, leading to rapid convergence to local optima, or too unbound, resulting in excessive exploration without convergence. As mentioned in introduction (lines 41-42), dynamically scheduling $k$ could achieve a *"balance between coarse-grained (larger $k$) and fine-grained (smaller $k$) searches"*. In our NeuOpt, $k$ is autonomously determined and dynamically scheduled during the search.

---

> > ### Comment · Reviewer_NeSp · 2023-08-18
> >
> > Thanks for clarifications and additional experiments. Despite limitations mentioned by [2dzf,vydT], I think that the algorithm is interesting. I will keep my original score.

---

> > > ### Author Response · Authors · 2023-08-19
> > > **Thank you for the support**
> > >
> > > We deeply appreciate the reviewer for acknowledging our response and additional experiments. Thank you for keeping to support our paper. We will incorporate the valuable suggestions from all the reviewers and ensure all limitations are addressed properly in the revised paper. Furthermore, we note that the Area Chair has kindly verified the receipt of our code repo link in a separate comment.

---

### Author Rebuttal · Authors · 2023-08-08

We thank all the reviewers for their valuable comments. We are pleased to see that the reviewers have recognized our NeuOpt approach (including the k-opt action factorization, the RDS decoder, the fresh constraint-handling scheme GIRE, and the dynamic data augmentation) as being **novel** (#NeSp and #8kc7), **general** (#8kc7), **of significant contribution** (#NeSp), **beneficial for use** (#vydT), and **reasonable** (#2dzf). We also appreciate the positive feedback, where all reviewers found our paper mostly **well-written** and our experiments with various baselines **extensive** and **comprehensive**. In this global response, we intend to address the common concerns.
***
**[Evaluation on more complex instances]** While Reviewers #NeSp and #8kc7 raised concerns regarding the evaluation of our approach on more complex (larger or real-world) instances, we would like to clarify that our NeuOpt (for TSP) and NeuOpt-GIRE (for CVRP) were indeed evaluated on these instances **in the original Appendix**. We will add summary tables and additional discussions to the revised main paper. Below we summarize the key results.
* **Results on larger instances:**
    * Firstly, we followed existing conventions to benchmark our approach against various baselines on TSP and CVRP instances with sizes 20, 50,100 in Table 1. Beyond these results that affirm the superior performance of our approach, we have also conducted experiments on larger TSP200 and CVRP200 instances as detailed in Appendix E.2 and Appendix Table 5. The results indicate that our approach consistently finds close-to-optimal solutions and still **outperforms the traditional LKH3 solver on CVRP200**. We note that due to prohibited longer training time, existing L2S solvers like DACT [1] may not be efficient for CVRP with size 200, highlighting the better scalability of our approach.
    * For solving even larger instances, we recommend integrating our NeuOpt with the SOTA **divide-and-conquer** frameworks (e.g., [2][3]) where they require efficient solvers for handling sub-problems with small size. Given the new SOTA performance on sizes 100 and 200, our NeuOpt could serve as a **more desirable conquer solver** to complement the learning-based divide solver in such frameworks. Besides, as outlined in Section 7, future work would focus on integrating our NeuOpt with predicted heatmaps by L2P solvers to reduce search space, employing more scalable encoders or efficient CUDA implementation to further improve the scalability.
    * Lastly, we also examined the **generalization of our model to larger sizes** (generalize models trained on size 100 to larger sizes), details of which are given in the next bullet point.
* **Results on real-world instances (generalization across size and distribution):**
    * In Appendix E.4 (Table 6 and Table 7), we have evaluated the generalization of our NeuOpt models trained on TSP and CVRP instances with size 100 and uniform distribution to real-world instances (from TSPLIB and CVRPLIB) with larger sizes (e.g., size 200) and/or different distributions (e.g., clustered node distributions, corner depot, much different demand settings, etc). For ease of reference, we have summarized the results in Table A1 and Table A2 in the attached PDF. The results consistently showcases that our NeuOpt could yield the lowest average gap.
    * Notably, the generalization of our NeuOpt is even more promising when compared to AMDKD method [4] which explicitly boosted the cross-distribution generalization performance of POMO based on knowledge distillation. This further suggests the promising potential of our NeuOpt if NeuOpt were to be augmented with a similar generalization boosting method in future work (e.g., applying AMDKD [4] or per-instance gradient update EAS [5] to our NeuOpt).
```
References:
[1] Learning to Iteratively Solve Routing Problems with Dual-Aspect Collaborative Transformer (NeurIPS’21)
[2] Learning to Delegate for Large-Scale Vehicle Routing (NeurIPS’21)
[3] RBG: Hierarchically Solving Large-Scale Routing Problems in Logistic Systems via Reinforcement Learning (KDD’22)
[4] Learning Generalizable Models for Vehicle Routing Problems via Knowledge Distillation (NuerIPS’22)
[5] Efficient active search for combinatorial optimization problems (ICLR’22)
```
***
**[Clarification on Figure 1]** We thank the reviewers for providing this feedback and we apologize for the confusion due to the space limitation. We will add more explanations in the revised paper and further simplify this figure. For clarification, Figure 1 depicts an example on TSP-9 to illustrate how our decoder determines a 3-opt exchange with K=4 steps. Even though K=4, the 3-opt is selected instead of the 4-opt due to the E-move being chosen at the final decoding step. The upper portion of the figure provides a visual representation of how the dual-stream attentions are computed, where the top yellow part represents the move stream $\mu$, while the bottom blue part represents the edge stream $\lambda$. The lower portion of the figure demonstrates how the inferred basis moves contribute to the modification of the current solution. At each step, $\kappa$, our RDS decoder computes the dual-stream attention (depicted by both yellow and blue dotted arrows) from representations of historical decisions $q^\kappa_\mu$, $q^\kappa_\lambda$ to node embeddings $h_i$. Each $q$ undergoes processing by the corresponding GRUs that model the historical decisions. Following the attention, one node is selected (highlighted in green), thereby deciding a basis move $\Phi_\kappa(x_\kappa)$ to modify the solution. Ghost marks are used to indicate the same location of a cyclic solution when viewed from a flat perspective as in this figure.
***
**[Additional experiments]** Following the suggestions of Reviewer #NeSp, #8kc7, and #vydT, we have added new experiments (figures and tables) in the attached PDF. Details can be found in our specific response to each reviewer.

---

### Decision · Program_Chairs · 2023-09-21

**Decision:**

Accept (poster)

**Comment:**

Reviewers agree that the paper is well-written, with clear and thorough experiments to demonstrate the effectiveness of proposed method (NeuOpt) that learns to search in combinatorial optimization problems. Therefore I vote for acceptance.